# GRAPH JOINT ATTENTION NETWORKS

## ABSTRACT

Graph attention networks (GATs) have been recognized as powerful tools for learning in graph structured data. However, how to enable the attention mechanisms in GATs to smoothly consider both structural and feature information is still very challenging. In this paper, we propose Graph Joint Attention Networks (JATs) to address the aforementioned challenge. Different from previous attention-based graph neural networks (GNNs), JATs adopt novel joint attention mechanisms which can automatically determine the relative significance between node features and structural coefficients learned from graph subspace, when computing the attention scores. Therefore, representations concerning more structural properties can be inferred by JATs. Besides, we theoretically analyze the expressive power of JATs and further propose an improved strategy for the joint attention mechanisms that enables JATs to reach the upper bound of expressive power which every message-passing GNN can ultimately achieve, i.e., 1-WL test. JATs can thereby be seen as most powerful message-passing GNNs. The proposed neural architecture has been extensively tested on widely used benchmarking datasets, including Cora, Cite, Pubmed, and OGBN-Arxiv, and has been compared with state-of-the-art GNNs for node classification tasks. Experimental results show that JATs achieve state-of-the-art performance on all the testing datasets.

## 1 INTRODUCTION

Many real-world data can be modeled as a graph, where a set of nodes (vertices), edges, and bag-of-words features respectively represent data instances, instance-instance interrelationships, and contents characterizing the nodes. For example, scientific articles in a research domain can be modeled as a graph, where nodes, edges, and node features respectively represent published articles, citations, and index information of the articles. Besides, social network users and interacted biological units can also be similarly represented as graphs possessing different structural and descriptive information. As graph data are widely available and they are related to various analytical tasks, learning in graphs has been a hot-spot in machine learning community.

There have been a number of approaches proposed to effectively learn in graph structured data. Amongst them, graph convolutional networks (GCNs) have shown to be powerful in learning low-dimensional representations for various subsequent analytical tasks. Different from those empirical convolutional neural networks (CNNs) which have achieved a great success in learning in image, vision, and natural language data (Krizhevsky et al., 2012; Xu et al., 2014), and whose convolution operators are always defined to process a grid-like data structure, GCNs attempt to formulate convolution operators aggregating the node features according to the observed graph structure, and learn the information propagation through different neural architectures. Meaningful representations which capture discriminative node features as well as intricate graph structure can thereby be learned by GCNs. There have been several sophisticated GCNs proposed in the recent. According to the ways through which GCNs make use of graph topology to define convolution operators for feature aggregation, GCNs can generally be categorized as spectral, and spatial ones (Wu et al., 2020).

Spectral GCNs define the convolutional layer for aggregating neighbor features based on the spectral representation of the graph. For example, Spectral CNN (Bruna et al., 2013) constructs the convolution layer based on the eigen-decomposition of graph Laplacian in the Fourier domain. However, such layer is computationally demanding. Aiming to reduce such computational burden, several approaches adopting the convolution operators which are based on simplified/approximate spectral graph theory are proposed. First, parameterized filters with smooth coefficients are introduced for

Spectral CNN to allow it to consider spatially localized nodes in the graph (Henaff et al., 2015). Chebyshev expansion is then introduced in (Defferrard et al., 2016) to approximate graph Laplacian rather than directly perform eigen-decomposition of it. Finally, the graph convolution filter is further simplified by only considering connected neighbors of each node (Kipf & Welling, 2017) so as to further make spectral GCNs computationally efficient.

In contrast, spatial GCNs define the convolution operators for feature aggregation directly making use of local structural properties of the central node. The key of spatial GCNs is consequently how to design an appropriate function for aggregating the effect brought by the features of candidate neighbors selected according to a proper sampling strategy. To achieve this, it sometimes requires to learn a weight matrix according to node degree (Duvenaud et al., 2015), utilize the power of transition matrix to preserve the neighbor importance (Atwood & Towsley, 2016; Busch et al., 2020; Klicpera et al., 2019), extract the normalized neighbors (Niepert et al., 2016), or sample a fixed number of neighbors (Hamilton et al., 2017; Zhang et al., 2020).

As a representative spatial GCN, Graph attention network (GAT) (Veličković et al., 2018; Zhang et al., 2020) has shown a promising performance in various graph learning tasks. What makes GATs effective in learning graph representations is they adopt the attention mechanism, which has been successfully used in machine reading and translation (Luong et al., 2015; Cheng et al., 2016), and video processing (Xu et al., 2015), to compute the node-feature-based attention weights (attention scores) between a central node and its one-hop neighbors (including the central node itself). Then, GATs utilize the attention scores to obtain a weighted aggregation of node features which are propagated to the next layer. As a result, those neighbors possessing similar features may impact more on the center node, and meaningful representations can be inferred by GATs.

Although GATs have been experimentally verified as powerful tools for various graph learning tasks, they still confront several challenges. First, for attention-based GNNs, appropriate attention mechanisms which can automatically identify the relative significance between the graph structure and node features are not many. As a result, most current attention mechanisms for GATs cannot effectively capture the joint effect brought by the underlying graph structure and node features for seamlessly impacting the message-passing in the neural architecture. Second, whether the expressive power of GNNs adopting the attention mechanisms which can effectively acquire the aforementioned joint effect may reach the upper bound of message-passing GNNs has not been theoretically investigated. To address the mentioned challenges, in this paper, we propose novel attention-based GNNs, dubbed Graph Joint Attention Networks (JATs). Different from previous works, the attention mechanisms adopted by JATs are able to automatically capture the relative significance between structural coefficient learned from graph topology, and node features, so that higher attention scores may be learned by those neighbors which are topologically and contextually correlated. JATs are consequently able to smoothly adjust attention scores according to the contemporary structure and node features, and truly capture the joint attention on structural and contextual information propagated in the neural network. Besides, we theoretically analyze the expressive power of JATs and further propose an improved strategy which enables JATs to distinguish all distinct graph structures as 1-dimensional Weisfeiler-Lehman test (1-WL test) does. This means JATs can reach the upper bound w.r.t. expressive power which all message-passing GNNs can ultimately achieve. JATs have been extensively tested on four widely used datasets, i.e., Cora, Citeseer, Pubmed, and OGBN-Arxiv, and have been compared with a number of strong baselines. The experimental results show that JATs achieve the state-of-the-art performance.

The rest of the paper is organized as follows. In Section 2, we elaborate the proposed JATs, and compare JATs with other GNNs. In Section 3, we prove the limitation w.r.t. expressive power of the joint attention mechanisms presented in Section 2. A strategy is then proposed to improve JATs to reach the upper bound of expressive power which all message-passing GNNs can at most achieve. The comprehensive experiments which are used to validate the effectiveness of JATs are presented in Section 4. Finally, we summarize the contributions of the paper and propose future works potentially improving JATs.

## 2 JOINT ATTENTION-BASED GRAPH NEURAL NETWORKS

In this section, we elaborate the proposed JATs. Mathematical preliminaries and notations used in the paper are firstly illustrated. How JATs learn the structural coefficients which are used in the

joint attention mechanisms is then introduced. Following that, the joint attention layer, which is the cornerstone of JATs is elaborated. At last, we compare the proposed JATs with their counterparts.

## 2.1 NOTATIONS AND PRELIMINARIES

Throughout this paper, we assume a graph $G = \{V, E\}$ containing $N$ nodes, $|E|$ edges, where $V$ and $E$ respectively represent the node and edge set. We use $\mathbf{A} \in \{0, 1\}^{N \times N}$ and $\mathbf{X} \in \mathbb{R}^{N \times D}$ to represent graph adjacency matrix and node feature matrix, respectively. $\mathcal{N}_i$ denotes the union of node $i$ and its one-hop neighbors. $\mathbf{W}^l$ and $\{\mathbf{h}_i^l\}_{i=1,...N}$ denote the weight matrix and features of node $i$ at $l$th layer of JATs, respectively, and $\mathbf{h}^0$ is set to be the input feature, i.e., $\mathbf{X}$. For the nodes in $\mathcal{N}_i$, their possible feature vectors form a multiset $M_i = (S_i, \mu_i)$, where $S_i = \{s_1, ...s_n\}$ is the ground set of $M_i$ which contains the distinct elements existing in $M_i$, and $\mu_i : S_i \to \mathbb{N}^\star$ is the multiplicity function indicating the frequency of occurrence of each distinct $s$ in $M_i$.

## 2.2 LEARNING STRUCTURAL COEFFICIENTS FROM GRAPH SUBSPACE

It is well known that topology is the corner stone of the graph. How to utilize such structural information to compute the attention scores naturally has a profound impact on the performance of attention-based GNNs. Empirical attention based GNNs, e.g., GAT, compute the attention scores between connected neighbors only using node features, but overlook the structural correlation between pairwise nodes. To allow attention-based GNNs to capture the higher-order structures in the graph, we propose JATs to learn the topological coefficients from the graph subspace. Inspired by subspace clustering (Elhamifar & Vidal, 2013), we may formulate the learning of structural coefficients as follows. Given $N$ nodes in the graph drawn from multiple linear subspaces $\{\mathcal{S}_i\}_{i=1,...K}$, one can represent a node in a subpace as a linear combination of other nodes. If each row in $\mathbf{A}$ is treated as the structural information of each node, one can simply represent it using other nodes (other rows in $\mathbf{A}$) as one equation, i.e., $\mathbf{A}_{i,:} = \mathbf{C}_{i,:}\mathbf{A}$, where $\mathbf{C}$ denotes the structural coefficient matrix as $\mathbf{A}$ is used. It has been shown in previous works (Ji et al., 2014) that under the assumption the subsapces are independent, by minimizing certain norm of $\mathbf{C}$, $\mathbf{C}$ may have a block diagonal structure (after finite permutations). In other words, $\mathbf{C}_{ij} \neq 0$ if and only if two nodes, $v_i$ and $v_j$ are in the same subspace. So, we can utilize $\mathbf{C}$ to learn the structural correlations between neighbors in the graph. And the above learning task can be formulated as the following optimization problem:

$$\text{minimize } \|\mathbf{C}\|_p, \text{subject to } \mathbf{A} = \mathbf{CA}, \mathbf{C}_{ii} = 0, \tag{1}$$

where $\|\cdot\|_p$ stands for a certain matrix norm, and the zero constraint on the diagonal of $\mathbf{C}$ may prevent trivial solutions when $\|\cdot\|_p$ is the norm considering sparsity. To make the data corruption explainable, the equality constraint in Eq. (1) is often relaxed as a regularization term and the learning of structural coefficients can be reformulated as follows:

$$\text{minimize } \|\mathbf{C}\|_p + \beta\|\mathbf{A} - \mathbf{CA}\|_F^2, \text{ subject to } \mathbf{C}_{ii} = 0. \tag{2}$$

By rewriting Eq. (2), we may reveal why subspace learning is effective in capturing the structural correlations between pairwise nodes. Mathematically, Eq. (2) is equivalent to the following node-wise minimization problem:

$$\text{minimize } \sum_i \|\mathbf{C}_{i,:}\|_p + \beta\|\mathbf{A}_{i,:} - \mathbf{CA}_{i,:}\|_F^2, \text{ subject to } \mathbf{C}_{ii} = 0. \tag{3}$$

As $\mathbf{CA}_{i,:}$ is equal to $\sum_j \mathbf{C}_{ij}\mathbf{A}_{j,:}$ and $\mathbf{C}_{ii} = 0$, one may easily find that minimizing Eq. (3) is equivalent to search for the optimal linear combination of other nodes that can be used to reconstruct the $i$th node in the graph. As a result, $\mathbf{C}_{ij}$ is high when $\mathbf{A}_{j,:}$, i.e., the global structure of $j$th node (rather than the local bias (Gong et al., 2018)), is similar to $\mathbf{A}_{i,:}$. By minimizing Eq. (2), one may identify those nodes which are in the same graph subspace, and the structural correlations between neighboring nodes can therefore be inferred directly. As the above learning problem can be effectively solved via gradient descent, JATs can optimize Eq. (2) together with the training of the neural architecture. Also, we use $l_1$ norm for $\mathbf{C}$ to force JATs to learn sparse structural coefficients.

## 2.3 JOINT ATTENTION LAYER

Having made the structural coefficients available, we are now presenting joint attention layer, which is the core module for building JATs and will be used in our experiments. Different from the attention layers utilized by other GNNs, the joint attention layer in the proposed framework adopts

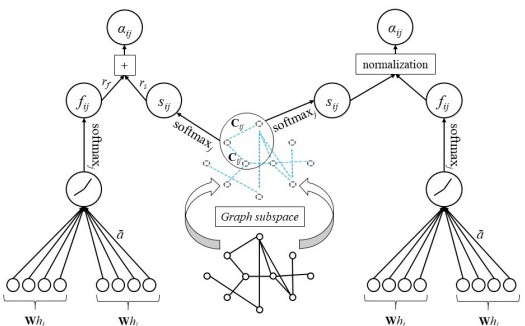

Figure 1: Graphical illustration of the joint attention mechanisms used in each layer of JATs. Left: Joint attention mechanism using *Implicit direction* strategy. Right: Joint attention mechanism using *Explicit direction* strategy. Both two mechanisms consider structural coefficients learned from graph adjacency.

novel attention mechanisms, which may automatically identify the relative significance between input features and structural information. Appropriate attention scores between central node and its neighbors can be obtained and meaningful representations can be inferred by JATs.

Given a set of node features $\{\mathbf{h}_i^l\}_{i=1,\dots N}$, each $\mathbf{h}_i^l \in R^{D^l}$, the joint attention layer of JATs is to map them into $D^{l+1}$ dimensional space $\{\mathbf{h}_i^{l+1}\}_{i=1,\dots N}$, according to the correlations of input features and graph topology. JAT first attempts to compute the contextual correlation between two connected nodes, say $v_i$ and $v_j$. To do so, we directly adopt the feature-based attention mechanism utilized by previous GATs (Veličković et al., 2018):

$$f_{ij} = \frac{\exp(\text{LeakyReLU}(\tilde{\mathbf{a}}^T(\mathbf{W}^l\mathbf{h}_i^l \parallel \mathbf{W}^l\mathbf{h}_j^l)))}{\sum_{k \in \mathcal{N}_i} \exp(\text{LeakyReLU}(\tilde{\mathbf{a}}^T(\mathbf{W}^l\mathbf{h}_i^l \parallel \mathbf{W}^l\mathbf{h}_k^l)))}, \tag{4}$$

where $\tilde{\mathbf{a}} \in \mathbb{R}^{2F^{l+1}}$ is a vector of parameters of the feedforward layer, $\parallel$ stands for the concatenation function, and $\mathbf{W}^l$ is $F^{l+1} \times F^l$ variable matrix for feature mapping. Based on Eq. (4), JAT may capture the feature correlations between connected nodes (first-order neighbors) by computing the similarities w.r.t. features mapped to next layer.

However, determining the attention scores solely based on node features leads a GNN to overlook the structural information hidden in the graph. To overcome this issue, we propose JAT to learn the structural coefficients as mentioned in Section 2.2 and utilize the novel joint attention mechanisms to appropriately compute the attention scores. Given learnable structural coefficient, $\mathbf{C}_{ij}$ between two nodes, JAT obtains the structural correlation as follows:

$$s_{ij} = \frac{\exp(\mathbf{C}_{ij})}{\sum_{k \in \mathcal{N}_i} \exp(\mathbf{C}_{ik})}. \tag{5}$$

Given $f_{ij}$ and $s_{ij}$, JAT has two strategies (attention mechanisms) to compute the final attention scores. The first mechanism is dubbed *Implicit direction*. It aims at computing the attention scores whose relative significance between structural and feature correlations can be automatically acquired. To do so, for each layer of the neural network, JAT introduces two learnable parameters, $g_f$ and $g_s$ to determine the relative significance between structure and feature correlations and it can be obtained as follows:

$$r_f = \frac{\exp(g_f)}{\exp(g_s) + \exp(g_f)}, r_s = \frac{\exp(g_s)}{\exp(g_s) + \exp(g_f)} \tag{6}$$

where $r_s$ or $r_f$ represents the normalized significance related to structure or node features. Given them, JAT is able to compute the attention score based on *Implicit direction* strategy:

$$\alpha_{ij} = \frac{r_f \cdot f_{ij} + r_s \cdot s_{ij}}{\sum_{k \in \mathcal{N}_i}[r_f \cdot f_{ik} + r_s \cdot s_{ik}]} = r_f \cdot f_{ij} + r_s \cdot s_{ij}. \tag{7}$$

Given the attention mechanism shown in Eq. (7), $\alpha_{ij}$ attempts to capture the weighted mean attention in terms of structural and feature correlations between neighbors. Compared with the attention

mechanism solely based on features of one-hop neighbors, $\alpha_{ij}$ computed by Eq. (7) may be softly adjusted according to the implicit impact brought by the structural coefficients. Moreover, the relative significance $r$ can also be automatically inferred by JAT as it is involved into the back propagation process. More smooth and appropriate attention scores can thereby be computed by JAT for learning meaningful representations.

To enhance the structural impact, JAT has another strategy, named as *Explicit direction*, to compute attention scores between neighbors. Given $f_{ij}$ and $s_{ij}$, the attention scores obtained via *Explicit direction* strategy is defined as follows:

$$\alpha_{ij} = \frac{f_{ij} \cdot s_{ij}}{\sum_{k \in \mathcal{N}_i} f_{ik} \cdot s_{ik}} \tag{8}$$

Compared with Eq. (7), the structural coefficient explicitly influences the magnitude of $f_{ij}$, so that those node pairs which are structurally irrelevant are impossible to assign with high attention weights. Utilizing *Explicit direction* strategy, JAT becomes more structure dependent when performing message passing in its neural architecture.

Having obtained the attention scores, JAT is now able to compute a linear combination of features corresponding to each node and its neighbors as the layer-wise output, which will be either propagated to the higher layer, or be used as the final representations for subsequent learning tasks. The mentioned output features can be computed as follows:

$$\mathbf{h}_i^{l+1} = \sum_{j \in \mathcal{N}_i} \alpha_{ij} \mathbf{W} \mathbf{h}_j^l. \tag{9}$$

In Fig. 1, the joint attention mechanisms proposed in this paper are graphically illustrated. And we may use a particular number of joint attention layers utilizing the proposed attention mechanisms to construct JATs. In practice, we also adopt the multi-head attention strategy (Vaswani et al., 2017) to stabilize the learning process. JATs either concatenate the node features from multiple hidden layers as the input for next layer, or compute the average of node features obtained by multiple units of output layers as the final node representations. The details on how to implement multi-head attention in graph neural networks can be checked in (Veličković et al., 2018).

## 2.4 COMPARISON TO RELATED GRAPH NEURAL NETWORKS

Based on the structure of the introduced joint attention layer, it is found that Graph Joint Attention Networks are quite different from previous related neural architectures and have several advantages.

When compared with spectral based GNNs, the proposed JATs provide a dynamic way to update layer-wise representations. In each layer, the updating function for node representation in prevalent spectral GNNs, can generally be formulated as $\mathbf{h}_i' = \sigma(\sum_{j \in \mathcal{N}_i} \widetilde{\mathbf{A}}_{ij} \mathbf{h}_j \mathbf{W})$, where $\widetilde{\mathbf{A}}$ represents a static matrix/tensor, which can be constructed according to graph normalization (Kipf & Welling, 2017), or graph diffusion (Klicpera et al., 2019; Busch et al., 2020), so as to preserve particular spectral properties of graph structure. As it shows, the message passing in spectral GNNs is defined according to the static matrix/tensor preserving structural properties. In contrast, JATs compute the attention score ($\alpha_{ij}$) for each pair of neighbors based on structural coefficient and node features, which naturally leads each layer in the neural network to pay more attention to those node pairs sharing similar structures and features. Higher-order representations concerning such analogous neighbors in the graph can therefore be learned by JATs.

When compared with spatial GNNs, especially those attention based ones (Veličković et al., 2018; Zhang et al., 2020), JATs adopt adaptive attention mechanisms which smoothly determine the relative significance between graph structure and node features, to compute more appropriate attention scores for representation learning. Compared with GATs (Veličković et al., 2018), JATs are able to compute attention scores for first order neighbors according to both structural coefficients and node features, so that the attention-based message passing considers more structural information. JATs are also different from those spatial GNNs which update node representations via sampling a fixed size of neighbors. For example, to reduce the computational demand, GraphSAGE (Hamilton et al., 2017) performs the task of graph representation learning via max pooling. Another attention based framework, ADSF (Zhang et al., 2020) also takes into consideration structural information when computing attention scores. However, ADSF cannot infer the relative significance between

the utilized topological information and node features, and ADSF considers the best $k$ neighbors for computing attention scores. Unlike JATs, such spatial GNNs sampling neighbors for representation updating cannot fully utilize the structural information provided by proximal nodes in the graph. Due to the flexibility of the proposed attention mechanisms, JATs can be readily combined with any paradigm which can learn the topological correlations, just like the structural coefficients used in this paper. This property enhances the applicability of the proposed JATs.

## 3 MORE EXPRESSIVELY POWERFUL JATS

Study on the expressive power of various GNNs has drawn much attention in the recent. It concerns whether a given GNN can distinguish different structures where vertices possessing various vectorized features. It has been found that what the neighborhood aggregation functions, or readout operators of all message-passing GNNs, including GCNs, GATs, and other related aim at are analogous to what 1-dimensional Weisfeiler-Lehman test (1-WL test), which is injective and iteratively operated in Weisfeiler-Lehman algorithm (Weisfeiler & Leman, 1968; Xu et al., 2018; Zhang & Xie, 2020), does. In other words, both aggregation/readout functions in message-passing GNNs and 1-WL test attempt to distinguish structures which are different in some ways. As a result, all message-passing GNNs are as most powerful as 1-WL test (Xu et al., 2018). The theoretical validation of the expressive power of a given GNN thereby lies in whether those adopted aggregation/readout functions are homogeneous to 1-WL test.

Having the effective framework evaluating the expressive power of a given message-passing GNN, one may naturally be interested in whether the proposed JATs can distinguish all different graph structures as 1-WL test does. As we mainly consider node classification tasks, in this section, we investigate the expressive power of the neighborhood aggregation function concerning the joint attention mechanisms in Eqs. (7) and (8). We firstly show that the neighborhood aggregation function utilized by JATs still fails to discriminate some graph structures possessing certain topological properties. Then, we propose a simple but effective strategy for the joint attention mechanisms to enable JATs to successfully distinguish all those graph structures that previously cannot be discriminated.

For the neighborhood aggregation utilizing the attention mechanism shown in Eq. (7), we have the following theorem pointing out the conditions under which the aggregation function fails to distinguish different structures.

**Theorem 1** *Assume the feature space $\mathcal{X}$ is countable and the aggregation function concerns the attention mechanism in Eq. (7) is represented as $h(c, X) = \sum_{x \in X} \alpha_{cx} g(x)$, where $c$ is the feature of center node, $X \in \mathcal{X}$ is a multiset containing the feature vectors from nodes in $\mathcal{N}_i$, $g(\cdot)$ is a function for mapping input feature $X$, and $\alpha_{cx}$ is the attention score between $f(c)$ and $f(x)$. For all $g$ and the joint attention mechanism in Eq. (7), $h(c_1, X_1) = h(c_2, X_2)$ if and only if $c_1 = c_2$, $X_1 = \{S, \mu_1\}$, $X_2 = \{S, \mu_2\}$, and $\sum_{y=x, y \in X_1} f_{c_1 y} - \sum_{y=x, y \in X_2} f_{c_2 y} = q[\sum_{y=x, y \in X_2} s_{c_2 y} - \sum_{y=x, y \in X_1} s_{c_1 y}]$, for $q = \frac{r_s}{r_f}$ and $x \in S$. In other words, $h$ will map different multiset into the same embedding iff the multisets have same central node feature, same underlying set, and the difference in feature-based attention scores is proportional ($\frac{r_s}{r_f}$) to the opposite of that in attention weights corresponding to structural coefficients.*

We leave the proof of all the theorems and corollaries in the appendix. For the aggregation function utilizing the attention mechanism shown in Eq. (8), we have the following theorem indicating the structures which cannot be correctly distinguished.

**Theorem 2** *Under the same assumptions shown in Theorem 1, for all $g$ and the joint attention mechanism in Eq. (8), $h(c_1, X_1) = h(c_2, X_2)$ if and only if $c_1 = c_2$, $X_1 = \{S, \mu_1\}$, $X_2 = \{S, \mu_2\}$, and $q \cdot \sum_{y=x, y \in X_1} \phi(\mathbf{C}_{c_1 x}) = \sum_{y=x, y \in X_2} \phi(\mathbf{C}_{c_2 y})$, for $q > 0$ and $x \in S$, where $\phi(\cdot)$ is an function for mapping values to $\mathbb{R}^+$. In other words, $h$ will map different multiset into the same embedding iff the multisets have same central node feature, same node features whose corresponding mapped structural coefficients are proportional.*

Theorems 1 and 2 indicate that the joint attention mechanisms may still fail to distinguish some graph structures, although the aggregation functions adopting the joint attention mechanisms may be more expressively powerful than empirical GATs. As node features and graph structure are

heterogeneous, intuitively, sub-structures satisfying the mentioned conditions should be infrequent. This may well explain why those attention-based GNNs concerning including structural factors into the computation of attention scores may experimentally perform better than GATs. However, when distinct multisets with corresponding structural properties meet the conditions mentioned in Theorems 1 and 2, the joint attention mechanisms in Eqs. (7) and (8) cannot correctly distinguish such multisets. Thus, JATs fail to reach the upper bound of expressive power of all message-passing GNNs, i.e., 1-WL test.

However, we are able to readily improve the expressive power of JATs to meet the condition of 1-WL test by slightly modifying the joint attention mechanisms. The modified attention mechanisms are defined as follows:

$$\alpha_{ij} = \begin{cases} \alpha_{ij} & j \in \mathcal{N}_i, j \neq i, \\ \alpha_{ij} + \epsilon \cdot \frac{1}{|\mathcal{N}_i|} & j = i, \epsilon > 0, \end{cases} \tag{10}$$

where $\alpha_{ij}$ is the attention weight obtained by either Eq. (7) or (8). Then, the newly obtained attention scores can be used to aggregate the node features passed to the higher layers. Next, we prove that such improved attention mechanisms reach the upper bound of message-passing GNNs via showing they can distinguish those structures possessing the properties mentioned in Theorems 1 and 2.

**Corollary 1** *Let $\mathcal{T}$ be the attention-based aggregator shown in Eq. (9) that considers the attention mechanism in Eq. (7) or (8) and operates on a multiset $H \in \mathcal{H}$, where $\mathcal{H}$ is a node feature space mapped from the countable input feature space $\mathcal{X}$. A $\mathcal{H}$ exists so that utilizing $\alpha_{ij}$ in Eq. (10), $\mathcal{T}$ can distinguish all different multisets in aggregation that it previously cannot discriminate.*

Based on Eq. (10), one may use original JATs to perform different classification tasks in graph data by setting $\epsilon = 0$. The expressive power of JATs can immediately reach the upper bound of message-passing GNNs when $\epsilon$ is set as a positive value. Theoretically, the expressive power of JATs is stronger than state-of-the-art attention-based GNNs, e.g., GATs (Veličković et al., 2018), and ADSF (Zhang et al., 2020). It should be noted that, all the theoretical validations presented in this paper assume the input feature space is countable. The general framework for validating GNNs' expressive power in the uncountable feature space can be checked in (Corso et al., 2020).

## 4 EXPERIMENTS AND ANALYSIS

In this section, we evaluate the proposed Graph Joint Attention Networks (JATs) against a variety of prevalent baselines, on widely used network datasets.

### 4.1 EXPERIMENTAL SET-UP

To validate the effectiveness of JATs, we compare them with a number of state-of-the-art baselines, including Gaussian fields and harmonic functions (GF) (Zhu et al., 2003), Manifold regularization (Mani-reg) (Belkin et al., 2006), Deepwalk (Perozzi et al., 2014), Semi-supervised graph embedding (Planetoid) (Yang et al., 2016), Node2Vec (Grover & Leskovec, 2016), Graph convolutional networks (GCN) (Kipf & Welling, 2017), GCN with Chebyshev filters (Chebyshev) (Defferrard et al., 2016), GraphSAGE (Hamilton et al., 2017), mixture model CNN (MoNet) (Monti et al., 2017), Graph attention networks (GAT) (Veličković et al., 2018), Bayesian GCN (BGCN) (Hasanzadeh et al., 2020), and Adaptive structural fingerprints (ADSF) (Zhang et al., 2020). Based on the experimental results previously reported, these baselines may represent the most advanced techniques for learning in graph structured data.

Four widely-used document networks, which are Cora, Citeseer, Pubmed (Lu & Getoor, 2003; Sen et al., 2008), and OGBN-Arxiv (Hu et al., 2020), are used in our experiments. Cora, Citeseer, and Pubmed are three classical network datasets for validating the effectiveness of GNNs. However, it is recently found that these three datasets sometimes may not effectively validate the predictive power of different graph learning approaches, due to the relatively small data size and data leakage (Shchur et al., 2018; Hu et al., 2020). Thus, more massive datasets having better data quality have been proposed to evaluate the performance of different approaches (Hu et al., 2020; Dwivedi et al., 2020). To effectively test JATs, we additionally use OGBN-Arxiv, which is available in the Open graph

Table 1: Average *Accuracy* on classical document networks

| Approaches | Cora | Citeseer | Pubmed |
|---|---|---|---|
| GF (Zhu et al., 2003) | 68.0% | 45.3% | 63.0% |
| Mani-reg (Belkin et al., 2006) | 59.5% | 60.1% | 70.7% |
| Deepwalk (Perozzi et al., 2014) | 67.2% | 43.2% | 65.3% |
| Planetoid (Yang et al., 2016) | 75.7% | 64.7% | 77.2% |
| Chebyshev (Defferrard et al., 2016) | 81.2% | 69.8% | 74.4% |
| GCN (Kipf & Welling, 2017) | 81.5% | 70.3% | 79.0% |
| MoNet (Monti et al., 2017) | 81.7% | – | 78.8% |
| GAT (Veličković et al., 2018) | 83.0% | 72.5% | 79.0% |
| BGCN Hasanzadeh et al. (2020) | 82.2% | 70.0% | – |
| ADSF (Zhang et al., 2020) | 84.7% | 73.8% | 79.4% |
| JAT-E | **85.5±0.4%** | **73.8±0.4%** | **82.0±0.3%** |
| JAT-I | **85.8±0.5%** | **74.3±0.4%** | **82.8±0.4%** |

Table 2: Average *Accuracy* on OGBN-Arxiv

| Approaches | Accuracy (%) | | |
|---|---|---|---|
| | Train | Validation | Test |
| MLP | 63.6% | 57.7% | 55.5% |
| Node2Vec (Grover & Leskovec, 2016) | 76.4% | 71.3% | 70.1% |
| GCN (Kipf & Welling, 2017) | 78.9% | 73.0% | 71.7% |
| GraphSAGE (Hamilton et al., 2017) | 82.3% | 72.8% | 71.5% |
| JAT-E | 81.3±0.2% | **73.8±0.1%** | 72.6±0.06% |
| JAT-I | 81.2±0.1% | 73.4±0.08% | **72.9±0.2%** |

benchmark database, as one of the testing datasets. The effectiveness of all methods is validated via allowing them to perform semi-supervised node classification (transductive learning) in all the benchmarking sets and the classified nodes are evaluated using $Accuracy$ $(Micro-F_1)$. To compare JATs impartially with other baselines, we closely follow the experimental paradigms used in the related works (Yang et al., 2016; Kipf & Welling, 2017; Veličković et al., 2018; Hu et al., 2020). We leave the details of testing datasets and experimental scenarios in appendix due to space limitation.

## 4.2 RESULTS ON NODE CLASSIFICATION

We obtain the average classification accuracy over 10 runs of JATs and compare it with other state-of-the-art approaches. The corresponding results are summarized in Table 1 and 2. As the tables show, JATs utilizing two different joint attention mechanisms outperform all the baselines on all the four testing datasets. Specifically, the average *Accuracy* on node classification obtained by JAT-I (JAT using *Implicit direction* strategy) achieves 85.8%, 74.3%, 82.8% and 72.9% on Cora, Citeseer, Pubmed, and OGBN-Arxiv, respectively. The presented experimental results demonstrate that the proposed method is one of the most effective GNNs for learning in graph structured data.

## 4.3 ABLATION STUDY

Besides testing the performance of transductive learning in network data, we further investigate how different settings of JATs may impact their performance. Specifically, we first investigate how JATs perform when different values of $\epsilon$ are used. The sensitivity test on $\epsilon = [10^{-8}, 10^{-6}, 10^{-4}, 10^{-2}, 10^{-1}, 1, 5, 10]$ is plotted in Fig. 2 (a) and (b). As the figures depict, both two versions of JATs perform robustly when $\epsilon \leq 1$ and the improved strategy shown in Eq. (10) may boost the performance of JATs in some datasets, e.g., Pubmed. Thus, we recommend to set $\epsilon \leq 1$ for both two versions of JATs. Such setting may ensure JATs leverage the joint attention mechanisms as well as preserve the expressive power, when classifying nodes in graph structured data.

Then, to further investigate whether the strategy of automatically determining the relative significance between feature and structural attention (Eqs. (6) and (7)) may truly improve the performance

Table 3: Performance comparison on JATs using different structural information. C, J, or SSL means JAT uses Cosine or Jaccard similarity to compute the structural correlations for attention score computation, or only uses structural coefficents to compute attention scores.

| Different versions of JATs | Cora | Citeseer | Pubmed | OGBN-Arxiv |
|---|---|---|---|---|
| JAT-EC | 85.3±0.1% | 73.4±0.3% | 80.0±0.1% | 72.4±0.2% |
| JAT-IC | 84.7±0.3% | 73.5±0.3% | 79.7±0.3% | 72.5±0.2% |
| JAT-EJ | 85.2±0.1% | 73.2±0.4% | 79.8±0.2% | **72.6±0.2%** |
| JAT-IJ | 85.1±0.2% | **73.8±0.1%** | 79.4±0.1% | 72.4±0.1% |
| JAT-SSL | **85.8 ±0.4%** | 73.1±0.7% | 81.0±0.6% | 72.4±0.07% |
| JAT-E | **85.5±0.4%** | **73.8±0.4%** | **82.0±0.3%** | **72.6±0.06%** |
| JAT-I | **85.8±0.5%** | **74.3±0.4%** | **82.8±0.4%** | **72.9+0.2%** |

on transductive learning, we compare JAT-I with GAT, ADSF, and JAT utilizing the attention strategy shown in Eq. (7) but setting $r_f = r_s = 0.5$ (JAT-I w/o ad). As Fig. 2 (c) shows, JAT-I can outperform all the baselines on the used testing datasets, including Cora, Citesser, and Pubmed. When compared with JAT-I w/o ad, JAT-I can significantly improve the performance on semi-supervised node classification on all the used datasets, at 95% confidence level. The strategy of automatic determination adopted by JATs is therefore effective in learning node representation for the subsequent predictive tasks.

Finally, we further test the effectiveness of the proposed joint attention mechanisms by replacing the structural coefficients (**C**) with other effective strategies. Specifically, we use Cosine and Jaccard similarity measures to compute the structural correlations for connected node pairs and then let JAT utilize them to compute attentions scores for the network training. Besides, we also allow one version of the JATs (JAT-SSL) to only use the structural coefficients learned via minimizing Eq. (2) to compute the attention scores in the training stage. The corresponding results are summarized in Table 3. As the table shows, JAT still performs robustly when using simple measures, e.g., Cosine and Jaccard similarity to obtain node-node structural correlations, although its performance is not better than the original versions of JAT. Given the results shown in Table 3, it is found that the proposed joint attention mechanisms are effective in learning node representations. The JAT framework can therefore be readily combined with many effective approaches which can infer node-node structural correlations, such as network embedding (Grover & Leskovec, 2016; Armandpour et al., 2019) and deep structural learning (Monti et al., 2017) for effective representation learning.

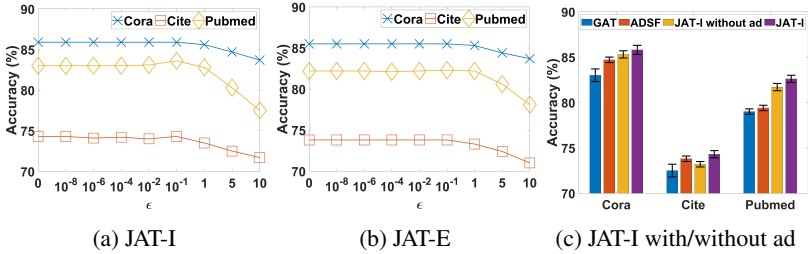

| (a) JAT-I | (b) JAT-E | (c) JAT-I with/without ad |
|---|---|---|

Figure 2: Sensitivity test on $\epsilon$ and automatic determination of S/F significance

## 5 CONCLUSION

In this paper, we propose novel attention-based GNNs, dubbed Graph Joint Attention Networks (JATs). Different from previous related approaches, JATs adopt novel joint attention mechanisms that can smoothly infer the relative significance between graph structure and node features, so that structural properties and node features are appropriately preserved in node representations. Besides, the expressive power of JATs is theoretically analyzed and the improved strategy to ensure JATs to be most powerful message-passing GNNs is also proposed. In future, we will further improve the effectiveness of JATs by considering different structural properties hidden in the network data and explore JATs' applicability by using them in multi-view and heterogeneous network data.

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

## A PROOF OF THEOREM 1

To prove Theorem 1, we need to consider the two directions of the iff conditions (Zhang & Xie, 2020). If given $c_1 = c_2$, $S_1 = S_2$, and $\sum_{y=x,y \in X_1} f_{c_1 y} - \sum_{y=x,y \in X_2} f_{c_2 y} = q[\sum_{y=x,y \in X_2} s_{c_2 y} - \sum_{y=x,y \in X_1} s_{c_1 y}]$, for $q = \frac{r_s}{r_f}$, for the aggregation function utilizing the joint attention mechanism shown in Eq. (7), we have:

$$
\begin{aligned}
h(c_i, X_i) &= \sum_{x \in X_i} \alpha_{c_i x} g(x), \\
\alpha_{c_i x} &= r_f \cdot f_{c_i x} + r_s \cdot s_{c_i x}, \\
f_{c_i x} &= \frac{\exp(m_{c_i x})}{\sum_{x \in X_i} \exp(m_{c_i x})}, s_{c_i x} = \frac{\exp(\mathbf{C}_{c_i x})}{\sum_{x \in X_i} \exp(\mathbf{C}_{c_i x})},
\end{aligned}
\tag{11}
$$

where $m_{c_i x}$ represents the feature similarity between $c_i$ and $x$. Given Eq. (11), we may directly derive $h(c_1, X_1)$ and $h(c_2, X_2)$:

$$
\begin{aligned}
h(c_1, X_1) &= \sum_{x \in X_1} \alpha_{c_1 x} g(x) = \sum_{x \in X_1} [r_f \cdot f_{c_1 x} + r_s \cdot s_{c_1 x}] \cdot g(x) \\
h(c_2, X_2) &= \sum_{x \in X_2} \alpha_{c_2 x} g(x) = \sum_{x \in X_2} [r_f \cdot f_{c_2 x} + r_s \cdot s_{c_2 x}] \cdot g(x)
\end{aligned}
\tag{12}
$$

Considering $c_1 = c_2$, $S_1 = S_2$, and $\sum_{y=x,y \in X_1} f_{c_1 y} - \sum_{y=x,y \in X_2} f_{c_2 y} = q[\sum_{y=x,y \in X_2} s_{c_2 y} - \sum_{y=x,y \in X_1} s_{c_1 y}]$, for $q = \frac{r_s}{r_f}$, we directly derive $h(c_1, X_1) = h(c_2, X_2)$.

If we are given $h(c_1, X_1) = h(c_2, X_2)$, we are able to prove that the conditions mentioned in the theorem are necessary by showing contradictions occur when they are not satisfied. If $h(c_1, X_1) = h(c_2, X_2)$, we have:

$$
\begin{aligned}
&h(c_1, X_1) - h(c_2, X_2) = \\
&\sum_{x \in X_1} [r_f \cdot f_{c_1 x} + r_s \cdot s_{c_1 x}] \cdot g(x) - \sum_{x \in X_2} [r_f \cdot f_{c_2 x} + r_s \cdot s_{c_2 x}] \cdot g(x) = 0
\end{aligned}
\tag{13}
$$

Firstly, assuming $S_1 \neq S_2$, for any $g(\cdot)$, we thereby have:

$$
\begin{aligned}
h(c_1, X_1) - h(c_2, X_2) &= \sum_{x \in S_1 \cap S_2} [\sum_{y=x,y \in X_1} [r_f \cdot f_{c_1 y} + r_s \cdot s_{c_1 y}] \\
&\quad - \sum_{y=x,y \in X_2} [r_f \cdot f_{c_2 y} + r_s \cdot s_{c_2 y}]] \cdot g(x) \\
&\quad + \sum_{x \in S_1 \setminus S_2} \sum_{y=x,y \in X_1} [r_f \cdot f_{c_1 y} + r_s \cdot s_{c_1 y}] \cdot g(x) \\
&\quad - \sum_{x \in S_2 \setminus S_1} \sum_{y=x,y \in X_2} [r_f \cdot f_{c_2 y} + r_s \cdot s_{c_2 y}] \cdot g(x) = 0
\end{aligned}
\tag{14}
$$

As Eq. (14) holds for any $g(\cdot)$, we may define another function $g'(\cdot)$ as follows:

$$
\begin{aligned}
g(x) &= g'(x), \text{for } x \in S_1 \cap S_2 \\
g(x) &= g'(x) - 1, \text{for } x \in S_1 \setminus S_2 \\
g(x) &= g'(x) + 1, \text{for } x \in S_2 \setminus S_1
\end{aligned}
\tag{15}
$$

If Eq. (14) holds, we also have:

$$
\begin{aligned}
h(c_1, X_1) - h(c_2, X_2) = &\sum_{x \in S_1 \cap S_2} [\sum_{y=x, y \in X_1} [r_f \cdot f_{c_1 y} + r_s \cdot s_{c_1 y}] \\
&- \sum_{y=x, y \in X_2} [r_f \cdot f_{c_2 y} + r_s \cdot s_{c_2 y}]] \cdot g'(x) \\
&+ \sum_{x \in S_1 \backslash S_2} \sum_{y=x, y \in X_1} [r_f \cdot f_{c_1 y} + r_s \cdot s_{c_1 y}] \cdot g'(x) \\
&- \sum_{x \in S_2 \backslash S_1} \sum_{y=x, y \in X_2} [r_f \cdot f_{c_2 y} + r_s \cdot s_{c_2 y}] \cdot g'(x) \\
= &\sum_{x \in S_1 \cap S_2} [\sum_{y=x, y \in X_1} [r_f \cdot f_{c_1 y} + r_s \cdot s_{c_1 y}] \\
&- \sum_{y=x, y \in X_2} [r_f \cdot f_{c_2 y} + r_s \cdot s_{c_2 y}]] \cdot g(x) \\
&+ \sum_{x \in S_1 \backslash S_2} \sum_{y=x, y \in X_1} [r_f \cdot f_{c_1 y} + r_s \cdot s_{c_1 y}] \cdot [g(x) - 1] \\
&- \sum_{x \in S_2 \backslash S_1} \sum_{y=x, y \in X_2} [r_f \cdot f_{c_2 y} + r_s \cdot s_{c_2 y}] \cdot [g(x) + 1] = 0
\end{aligned}
\tag{16}
$$

As Eq. (14) equals Eq. (16), we have:

$$
\sum_{x \in S_1 \backslash S_2} \sum_{y=x, y \in X_1} [r_f \cdot f_{c_1 y} + r_s \cdot s_{c_1 y}] + \sum_{x \in S_2 \backslash S_1} \sum_{y=x, y \in X_2} [r_f \cdot f_{c_2 y} + r_s \cdot s_{c_2 y}] = 0
\tag{17}
$$

Obviously, the above equation does not hold as the terms in the summation operator are positive. Thus, $S_1 \neq S_2$ is not true. We may now assume $S_1 = S_2 = S$. Eliminating the irrational terms in Eq. (14), we have:

$$
\sum_{x \in S_1 \cap S_2} [\sum_{y=x, y \in X_1} [r_f \cdot f_{c_1 y} + r_s \cdot s_{c_1 y}] - \sum_{y=x, y \in X_2} [r_f \cdot f_{c_2 y} + r_s \cdot s_{c_2 y}]] \cdot g(x) = 0
\tag{18}
$$

Thus we know each term in the summation equals zero:

$$
\sum_{y=x, y \in X_1} [r_f \cdot f_{c_1 y} + r_s \cdot s_{c_1 y}] - \sum_{y=x, y \in X_2} [r_f \cdot f_{c_2 y} + r_s \cdot s_{c_2 y}] = 0
\tag{19}
$$

As structural coefficients and node feature similarity are heterogeneous, we may assume $\sum_{y=x, y \in X_1} r_s \cdot s_{c_1 y} = \sum_{y=x, y \in X_2} r_s \cdot s_{c_2 y}$. Eq. (19) can be simplified and rewritten as:

$$
\frac{\mu_1(x)}{\mu_2(x)} = \frac{\exp(m_{c_2 x}) \sum_{x \in S_1} \sum_{y=x, y \in X_1} \exp(m_{c_1 x})}{\exp(m_{c_1 x}) \sum_{x \in S_2} \sum_{y=x, y \in X_2} \exp(m_{c_2 x})}
\tag{20}
$$

It is obvious that LHS of Eq. (20) is a rational number. However, the RHS of Eq. (20) can be an irrational number. We may consider $S = \{s, s_0\}$ and assume $c_1 = s_0$, $c_2 = s$. We may also assume the feature similarity between central node and others as follows:

$$
\begin{aligned}
m_{c_1 x} &= 1, \text{for } x \in S \\
m_{c_2 s} &= 1, m_{c_2 s_0} = 2
\end{aligned}
\tag{21}
$$

Consider $x = s$, we have:

$$
\frac{\mu_1(s)}{\mu_2(s)} = \frac{|X_1|}{|X_2| - n + ne},
\tag{22}
$$

where $n$ stands for the number of $s_0$ in $X_2$. It is obvious that the above equality does not hold as the RHS is an irrational number, while LHS is a rational number. Thus $c_1 \neq c_2$ is false. Given $c_1 = c_2$, Eq. (19) can be rewritten as:

$$
\sum_{y=x, y \in X_1} r_f \cdot f_{c_1 y} - \sum_{y=x, y \in X_2} r_f \cdot f_{c_2 y} + \sum_{y=x, y \in X_1} r_s \cdot s_{c_1 y} - \sum_{y=x, y \in X_2} r_s \cdot s_{c_2 y} = 0
\tag{23}
$$

To ensure above equation holds, we have:

$$\sum_{y=x,y\in X_1} f_{c_1y} - \sum_{y=x,y\in X_2} f_{c_2y} = \frac{r_s}{r_f}[\sum_{y=x,y\in X_2} s_{c_2y} - \sum_{y=x,y\in X_1} s_{c_1y}] \tag{24}$$

Further denoting $\frac{r_s}{r_f} = q$, we have $\sum_{y=x,y\in X_1} f_{c_1y} - \sum_{y=x,y\in X_2} f_{c_2y} = q[\sum_{y=x,y\in X_2} s_{c_2y} - \sum_{y=x,y\in X_1} s_{c_1y}]$. $\quad\square$

## B  PROOF OF THEOREM 2

To prove Theorem 2, we can follow the procedure which is used to prove Theorem 1. If given $c_1 = c_2$, $S_1 = S_2$, and $q \cdot \sum_{y=x,y\in X_1} \phi(\mathbf{C}_{c_1y}) = \sum_{y=x,y\in X_1} \phi(\mathbf{C}_{c_2y})$, for $q > 0$, we may directly replace $\phi(\cdot)$ using $\exp\{\cdot\}$. For the aggregation function using the joint attention mechanism in Eq. (8) we have:

$$h(c_i, X_i) = \sum_{x\in X_i} \alpha_{c_ix}g(x),$$
$$\alpha_{c_ix} = \frac{f_{c_ix} \cdot s_{c_ix}}{\sum_{x\in X_i} f_{c_ix} \cdot s_{c_ix}}, \tag{25}$$
$$f_{c_ix} = \frac{\exp(m_{c_ix})}{\sum_{x\in X_i} \exp(m_{c_ix})}, s_{c_ix} = \frac{\exp(\mathbf{C}_{c_ix})}{\sum_{x\in X_i} \exp(\mathbf{C}_{c_ix})},$$

where $m_{c_ix}$ represents the feature similarity between $c_i$ and $x$. Given Eq. (25), we may directly derive $h(c_1, X_1)$ and $h(c_2, X_2)$:

$$h(c_1, X_1) = \sum_{x\in X_1} \alpha_{c_1x}g(x) = \sum_{x\in X_1} [\frac{f_{c_1x} \cdot s_{c_1x}}{\sum_{x\in X_1} f_{c_1x} \cdot s_{c_1x}}] \cdot g(x)$$
$$h(c_2, X_2) = \sum_{x\in X_2} \alpha_{c_2x}g(x) = \sum_{x\in X_2} [\frac{f_{c_2x} \cdot s_{c_2x}}{\sum_{x\in X_2} f_{c_2x} \cdot s_{c_2x}}] \cdot g(x) \tag{26}$$

Given $S_1 = S_2$ and $c_1 = c_2$, $h(c_2, X_2)$ can be rewritten as:

$$h(c_2, X_2) = \sum_{x\in S_2} [\frac{f_{c_2x} \cdot \sum_{y=x,y\in X_2} s_{c_2y}}{\sum_{x\in S_2} f_{c_2x} \cdot \sum_{y=x,y\in X_2} s_{c_2y}}] \cdot g(x)$$
$$= \sum_{x\in S_2} [\frac{\frac{\exp(m_{c_2x})}{\sum_{y\in X_2} \exp(m_{c_2y})} \cdot \sum_{y=x,y\in X_2} \frac{\exp(\mathbf{C}_{c_2y})}{\sum_{y\in X_2} \exp(\mathbf{C}_{c_2y})}}{\sum_{x\in S_2} \frac{\exp(m_{c_2x})}{\sum_{y\in X_2} \exp(m_{c_2y})} \sum_{y=x,y\in X_2} \frac{\exp(\mathbf{C}_{c_2y})}{\sum_{y\in X_2} \exp(\mathbf{C}_{c_2y})}}] \cdot g(x) \tag{27}$$
$$= \sum_{x\in S_2} \frac{\exp(m_{c_2x}) \sum_{y=x,y\in X_2} \exp(\mathbf{C}_{c_2y})}{\sum_{x\in S_2} \exp(m_{c_2x}) \sum_{y=x,y\in X_2} \exp(\mathbf{C}_{c_2y})} \cdot g(x)$$

Given $q \cdot \sum_{x\in X_1} \exp\{\mathbf{C}_{c_1x}\} = \sum_{y=x} \exp\{\mathbf{C}_{c_2y}\}$, Eq. (27) is equivalent to:

$$h(c_2, X_2) = \sum_{x\in S_1} \frac{q \cdot \exp(m_{c_2x}) \sum_{y=x,y\in X_1} \exp(\mathbf{C}_{c_1y})}{\sum_{x\in S_1} q \cdot \exp(m_{c_2x}) \sum_{y=x,y\in X_1} \exp(\mathbf{C}_{c_1y})} \cdot g(x) \tag{28}$$

Considering $c_1 = c_2$, we have $h(c_1, X_1) = h(c_2, X_2)$.

If we are given $h(c_1, X_1) = h(c_2, X_2)$, we are able to prove the conditions mentioned in the theorem are necessary by showing contradictions occur when they are not satisfied. If $h(c_1, X_1) = h(c_2, X_2)$, we have:

$$h(c_1, X_1) - h(c_2, X_2) =$$
$$\sum_{x\in X_1} [\frac{f_{c_1x} \cdot s_{c_1x}}{\sum_{x\in X_1} f_{c_1x} \cdot s_{c_1x}}] \cdot g(x) - \sum_{x\in X_2} [\frac{f_{c_2x} \cdot s_{c_2x}}{\sum_{x\in X_2} f_{c_2x} \cdot s_{c_2x}}] \cdot g(x) = 0 \tag{29}$$

Firstly, assuming $S_1 \neq S_2$, for any $g(\cdot)$, we thereby have:

$$
\begin{aligned}
h(c_1, X_1) - h(c_2, X_2) = & \sum_{x \in S_1 \cap S_2} \Big[\frac{\exp\left(m_{c_1 x}\right) \sum_{y=x, y \in X_1} \exp\left(\mathbf{C}_{c_1 y}\right)}{\sum_{x \in S_1} \exp\left(m_{c_1 x}\right) \sum_{y=x, y \in X_1} \exp\left(\mathbf{C}_{c_1 y}\right)} \\
& - \frac{\exp\left(m_{c_2 x}\right) \sum_{y=x, y \in X_2} \exp\left(\mathbf{C}_{c_2 y}\right)}{\sum_{x \in S_2} \exp\left(m_{c_2 x}\right) \sum_{y=x, y \in X_2} \exp\left(\mathbf{C}_{c_2 y}\right)}\Big] \cdot g(x) \\
& + \sum_{x \in S_1 \backslash S_2} \Big[\frac{\exp\left(m_{c_1 x}\right) \sum_{y=x, y \in X_1} \exp\left(\mathbf{C}_{c_1 y}\right)}{\sum_{x \in S_1} \exp\left(m_{c_1 x}\right) \sum_{y=x, y \in X_1} \exp\left(\mathbf{C}_{c_1 y}\right)}\Big] \cdot g(x) \\
& - \sum_{x \in S_2 \backslash S_1} \Big[\frac{\exp\left(m_{c_2 x}\right) \sum_{y=x, y \in X_2} \exp\left(\mathbf{C}_{c_2 y}\right)}{\sum_{x \in S_2} \exp\left(m_{c_2 x}\right) \sum_{y=x, y \in X_2} \exp\left(\mathbf{C}_{c_2 y}\right)}\Big] \cdot g(x) = 0
\end{aligned}
\tag{30}
$$

As Eq. (30) holds for any $g(\cdot)$, we may define another function $g'(\cdot)$ as shown in Eq. (15). If Eq. (30) holds, we also have:

$$
\begin{aligned}
h(c_1, X_1) - h(c_2, X_2) = & \sum_{x \in S_1 \cap S_2} \Big[\frac{\exp\left(m_{c_1 x}\right) \sum_{y=x, y \in X_1} \exp\left(\mathbf{C}_{c_1 y}\right)}{\sum_{x \in S_1} \exp\left(m_{c_1 x}\right) \sum_{y=x, y \in X_1} \exp\left(\mathbf{C}_{c_1 y}\right)} \\
& - \frac{\exp\left(m_{c_2 x}\right) \sum_{y=x, y \in X_2} \exp\left(\mathbf{C}_{c_2 y}\right)}{\sum_{x \in S_2} \exp\left(m_{c_2 x}\right) \sum_{y=x, y \in X_2} \exp\left(\mathbf{C}_{c_2 y}\right)}\Big] \cdot g'(x) \\
& + \sum_{x \in S_1 \backslash S_2} \Big[\frac{\exp\left(m_{c_1 x}\right) \sum_{y=x, y \in X_1} \exp\left(\mathbf{C}_{c_1 y}\right)}{\sum_{x \in S_1} \exp\left(m_{c_1 x}\right) \sum_{y=x, y \in X_1} \exp\left(\mathbf{C}_{c_1 y}\right)}\Big] \cdot g'(x) \\
& - \sum_{x \in S_2 \backslash S_1} \Big[\frac{\exp\left(m_{c_2 x}\right) \sum_{y=x, y \in X_2} \exp\left(\mathbf{C}_{c_2 y}\right)}{\sum_{x \in S_2} \exp\left(m_{c_2 x}\right) \sum_{y=x, y \in X_2} \exp\left(\mathbf{C}_{c_2 y}\right)}\Big] \cdot g'(x) \\
= & \sum_{x \in S_1 \cap S_2} \Big[\frac{\exp\left(m_{c_1 x}\right) \sum_{y=x, y \in X_1} \exp\left(\mathbf{C}_{c_1 y}\right)}{\sum_{x \in S_1} \exp\left(m_{c_1 x}\right) \sum_{y=x, y \in X_1} \exp\left(\mathbf{C}_{c_1 y}\right)} \\
& - \frac{\exp\left(m_{c_2 x}\right) \sum_{y=x, y \in X_2} \exp\left(\mathbf{C}_{c_2 y}\right)}{\sum_{x \in S_2} \exp\left(m_{c_2 x}\right) \sum_{y=x, y \in X_2} \exp\left(\mathbf{C}_{c_2 y}\right)}\Big] \cdot g(x) \\
& + \sum_{x \in S_1 \backslash S_2} \Big[\frac{\exp\left(m_{c_1 x}\right) \sum_{y=x, y \in X_1} \exp\left(\mathbf{C}_{c_1 y}\right)}{\sum_{x \in S_1} \exp\left(m_{c_1 x}\right) \sum_{y=x, y \in X_1} \exp\left(\mathbf{C}_{c_1 y}\right)}\Big] \cdot [g(x) - 1] \\
& - \sum_{x \in S_2 \backslash S_1} \Big[\frac{\exp\left(m_{c_2 x}\right) \sum_{y=x, y \in X_2} \exp\left(\mathbf{C}_{c_2 y}\right)}{\sum_{x \in S_2} \exp\left(m_{c_2 x}\right) \sum_{y=x, y \in X_2} \exp\left(\mathbf{C}_{c_2 y}\right)}\Big] \cdot [g(x) + 1] = 0
\end{aligned}
\tag{31}
$$

As Eq. (30) equals Eq. (31), we have:

$$
\begin{aligned}
& \sum_{x \in S_1 \backslash S_2} \Big[\frac{\exp\left(m_{c_1 x}\right) \sum_{y=x, y \in X_1} \exp\left(\mathbf{C}_{c_1 y}\right)}{\sum_{x \in S_1} \exp\left(m_{c_1 x}\right) \sum_{y=x, y \in X_1} \exp\left(\mathbf{C}_{c_1 y}\right)}\Big] \\
& + \sum_{x \in S_2 \backslash S_1} \Big[\frac{\exp\left(m_{c_2 x}\right) \sum_{y=x, y \in X_2} \exp\left(\mathbf{C}_{c_2 y}\right)}{\sum_{x \in S_2} \exp\left(m_{c_2 x}\right) \sum_{y=x, y \in X_2} \exp\left(\mathbf{C}_{c_2 y}\right)}\Big] = 0
\end{aligned}
\tag{32}
$$

Obviously, the above equation does not hold as softmax function is positive. Thus, $S_1 \neq S_2$ is not true. We may now assume $S_1 = S_2 = S$. Eliminating the irrational terms in Eq. (30), we have:

$$
\begin{aligned}
& \sum_{x \in S_1 \cap S_2} \Big[\frac{\exp\left(m_{c_1 x}\right) \sum_{y=x, y \in X_1} \exp\left(\mathbf{C}_{c_1 y}\right)}{\sum_{x \in S_1} \exp\left(m_{c_1 x}\right) \sum_{y=x, y \in X_1} \exp\left(\mathbf{C}_{c_1 y}\right)} \\
& - \frac{\exp\left(m_{c_2 x}\right) \sum_{y=x, y \in X_2} \exp\left(\mathbf{C}_{c_2 y}\right)}{\sum_{x \in S_2} \exp\left(m_{c_2 x}\right) \sum_{y=x, y \in X_2} \exp\left(\mathbf{C}_{c_2 y}\right)}\Big] \cdot g(x) = 0
\end{aligned}
\tag{33}
$$

Thus we know each term in the summation equals zero:

$$\frac{\exp\left(m_{c_1 x}\right)\sum_{y=x,y\in X_1}\exp\left(\mathbf{C}_{c_1 y}\right)}{\sum_{x\in S_1}\exp\left(m_{c_1 x}\right)\sum_{y=x,y\in X_1}\exp\left(\mathbf{C}_{c_1 y}\right)} - \frac{\exp\left(m_{c_2 x}\right)\sum_{y=x,y\in X_2}\exp\left(\mathbf{C}_{c_2 y}\right)}{\sum_{x\in S_2}\exp\left(m_{c_2 x}\right)\sum_{y=x,y\in X_2}\exp\left(\mathbf{C}_{c_2 y}\right)} = 0 \tag{34}$$

Eq. (34) is equivalent to:

$$\frac{\sum_{y=x,y\in X_1}\exp\left(\mathbf{C}_{c_1 y}\right)}{\sum_{y=x,y\in X_2}\exp\left(\mathbf{C}_{c_2 y}\right)} = \frac{\exp\left(m_{c_2 x}\right)\sum_{x\in S_1}\exp\left(m_{c_1 x}\right)\sum_{y=x,y\in X_1}\exp\left(\mathbf{C}_{c_1 y}\right)}{\exp\left(m_{c_1 x}\right)\sum_{x\in S_2}\exp\left(m_{c_2 x}\right)\sum_{y=x,y\in X_2}\exp\left(\mathbf{C}_{c_2 y}\right)} \tag{35}$$

We may consider $S = \{s, s_0\}$ and assume $c_1 = s_0$, $c_2 = s$. We may also assume the feature similarity between central node and others as follows:

$$m_{c_1 x} = 1, \text{for } x \in S$$
$$m_{c_2 s} = 1, m_{c_2 s_0} = 2 \tag{36}$$

Consider $x = s$, we have:

$$\frac{\sum_{s\in X_1}\exp\left(\mathbf{C}_{c_1 s}\right)}{\sum_{s\in X_2}\exp\left(\mathbf{C}_{c_2 s}\right)} = \frac{e[e\sum_{s\in X_1}\exp\left(\mathbf{C}_{c_1 s}\right) + e\sum_{s_0\in X_1}\exp\left(\mathbf{C}_{c_1 s_0}\right)]}{e[e\sum_{s\in X_2}\exp\left(\mathbf{C}_{c_2 s}\right) + e^2\sum_{s_0\in X_2}\exp\left(\mathbf{C}_{c_2 s_0}\right)]} \tag{37}$$

As the learning of $\mathbf{C}$ is independent of feature mapping, and the computation of attention coefficients, $\exp\left(\mathbf{C}_{cx}\right)$ can be any positive value. By setting the exponential values in the above equation as $a$, which is a positive value. We have $\frac{\mu_1(s)}{\mu_2(s)} = \frac{|X_1|}{|X_2|-n+ne}$. Similar with Eq. (22), $c_1 \neq c_2$ is not true. Since $c_1 = c_2 = c$, Eq. (35) can be rewritten as:

$$\frac{\sum_{y=x,y\in X_1}\exp\left(\mathbf{C}_{cy}\right)}{\sum_{y=x,y\in X_2}\exp\left(\mathbf{C}_{cy}\right)} = \frac{\sum_{x\in S_1}\exp\left(m_{cx}\right)\sum_{y=x,y\in X_1}\exp\left(\mathbf{C}_{cy}\right)}{\sum_{x\in S_2}\exp\left(m_{cx}\right)\sum_{y=x,y\in X_2}\exp\left(\mathbf{C}_{cy}\right)} = const > 0. \tag{38}$$

By setting $const$ as $\frac{1}{q}$ and $\exp\left(\mathbf{C}_{cy}\right) = \phi(\mathbf{C}_{cy})$, we have $q\sum_{y=x,y\in X_1}\phi(\mathbf{C}_{cy}) = \sum_{y=x,y\in X_2}\phi(\mathbf{C}_{cy})$. □

## C   PROOF OF COROLLARY 1

For any two distinct multisets $X_1$ and $X_2$ that cannot be distinguished by $\mathcal{T}$, according to **Theorem 1**, we denote $X_1 = (S, \mu_1)$, $X_2 = (S, \mu_2)$, $c \in S$. We also assume $\sum_{y=x,y\in X_1} f_{c_1 y} - \sum_{y=x,y\in X_2} f_{c_2 y} = q[\sum_{y=x,y\in X_2} s_{c_2 y} - \sum_{y=x,y\in X_1} s_{c_1 y}]$, for $q = \frac{r_s}{r_f}$. When using the attention mechanism in Eq. (7) for the aggregation function, it is easy to verify $\sum_{x\in X_1}\alpha_{cx}f(x) = \sum_{x\in X_2}\alpha_{cx}f(x)$. When using the improved mechanism shown in Eq. (10), we have $\sum_{x\in X_1}\alpha_{cx}f(x) - \sum_{x\in X_2}\alpha_{cx}f(x) = \epsilon(\frac{1}{|X_1|} - \frac{1}{|X_2|})\alpha_{cc}f(c)$, where $|X_1| = |\mathcal{N}_1|$, and $|X_2| = |\mathcal{N}_2|$. Since $|X_1| \neq |X_2|$, $\sum_{x\in X_1}\alpha_{cx}f(x) - \sum_{x\in X_2}\alpha_{cx}f(x) \neq 0$, meaning the improved joint attention mechanism can distinguish those distinct multiset which $\mathcal{T}$ fails to distinguish. Similarly, we can prove the aggregation function using the improved attention mechanism concerning Eq. (8) also can distinguish those distinct multisets which $\mathcal{T}$ fails to distinguish. □

## D   DETAILED EXPERIMENTAL SET-UP AND ADDITIONAL TESTS

In this section, how the experiments used to validate the effectiveness of the proposed JATs are set up is introduced with more details.

### D.1   DATASET DESCRIPTION

Four document networks, which are Cora, Citeseer, Pubmed (Lu & Getoor, 2003; Sen et al., 2008), and OGBN-Arxiv (Hu et al., 2020), are used in our experiments to validate the effectiveness of different GNNs. In these four networks, vertices, edges, and vertex features represent the documents, citations between pairwise documents, and the bag-of-words representations of the documents, respectively. Besides, each node in these four document networks has a class label.

Table 4: Characteristics of the testing datasets used in our experiments

|  | **Cora** | **Citeseer** | **Pubmed** | **OGBN-Arxiv** |
|---|---|---|---|---|
| $N$ | 2708 | 3327 | 19717 | 169343 |
| $|E|$ | 5429 | 4732 | 44338 | 1166243 |
| $D$ | 1433 | 3703 | 500 | 128 |
| $C$ | 7 | 6 | 3 | 40 |
| **Training Nodes** | 140 | 120 | 60 | 90941 |
| **Validation Nodes** | 500 | 500 | 500 | 29799 |
| **Test Nodes** | 1000 | 1000 | 1000 | 48603 |

As graph transductive learning is mainly considered in our experiment to validate different approaches, we set up the experiment closely following the settings in the related works (Yang et al., 2016; Veličković et al., 2018; Hu et al., 2020). All the datasets are split into three parts: training, validation, and testing. For datasets Cora, Citeseer, and Pubmed, we use only 20 nodes per class for training, but all the feature vectors. For dataset OGBN-Arxiv, we use a practical split strategy, segmenting the nodes which represent the academic papers, according to publication years (Hu et al., 2020). Papers published up to 2017 are in the training split. While papers published in 2018 and 2019 are used for validation and test, respectively. The performance of different approaches are evaluated on the test split. The statistics of these benchmarking datasets are summarized in Table 4, where $N$, $|E|$, $D$, and $C$ denote the number of vertices, edges, vertex features, and number of classes in each dataset, respectively.

### D.2 EXPERIMENTAL SCENARIOS

The effectiveness of different methods is validated via allowing them to perform semi-supervised learning (transductive learning) in the aforementioned graph datasets. To compare the proposed JATs impartially with other baselines, we closely follow the experimental paradigms used in previous works (Yang et al., 2016; Kipf & Welling, 2017; Veličković et al., 2018; Hu et al., 2020). Specifically, only partial nodes in each class are labeled in the training stage, and all the graph data, including vertex features and edges, can be accessible for the training of neural networks. The predictive ability of the approaches is validated using the nodes in the test split. Algorithm performance is evaluated using $Accuracy$ ($Micro - F_1$).

For the network structure of JAT, following GAT (Veličković et al., 2018), we use a similar two-layer deep neural network to learn node representations in all the testing datasets. The first layer consists of multiple attention heads computing multiple features, followed by an exponential linear unit. In our experiment, eight attention heads are used for Cora, Citeseer, and Pubmed datasets and three attention heads are used for OGBN-Arxiv. The second layer, composed of attention heads and a softmax activation is used for node classification. For Cora, Citeseer and OGBN-Arxiv, a single attention head is used, while for Pubmed, eight output attention heads are used, to compute $C$ features. As for the dimensionality of the hidden units, we also following the settings used in previous works (Veličković et al., 2018; Hu et al., 2020). The dimension of hidden units is set as 8 for Cora, Cite, and Pubmed, and it is set as 256 for OGBN-Arxiv. Regularization is applied within the model to cope with the small training set sizes. During training, $L_2$ regularization with $\lambda = 0.0005$ and dropout with $p = 0.6$ are applied for both layers' inputs when JAT is training with Cora, Citeseer, and Pubmed. In Ogbn-Arxiv dataset, the setting of $\lambda = 0$ and dropout with $p = 0.05$ is used.

All the networks are initialized using Glorot initialization and trained to minimize cross-entropy on the training nodes using the Adam SGD optimizer (Kingma & Ba, 2014). A learning rate of 0.01 is used for datasets Cora, Citeseer, and Pubmed, while that of 0.002 is used in dataset OGBN-Arxiv. As JAT requires structural coefficients to compute attention scores, Eq. (2) needs optimizing. In our experiments, we use two different strategies to obtain optimal structural coefficients. For those datasets with small size, e.g., Cora, Citeseer, and Pubmed, Eq. (2) is optimized together with the training of the networks. In other words, a loss function aggregating cross-entropy on the training nodes and structure reconstruction error Eq. (2) is used. Eq. (2) is iteratively optimized along with the training of JAT and the structural coefficients obtained in each epoch can be used by JAT for

the computation of attention scores. For datasets possessing huge number of nodes, e.g., OGBN-Arxiv, Eq. (2) is optimized ahead of training JATs for the purpose of dealing with limited computing resources. Under such a case, the training efficiency of JATs can be significantly improved.

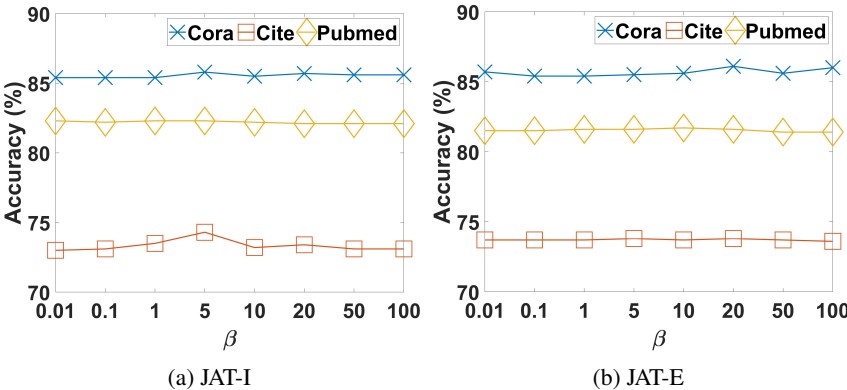

(a) JAT-I            (b) JAT-E

Figure 3: Sensitivity test on $\beta$

## D.3 SENSITIVITY TEST ON $\beta$

For sensitivity test on $\beta$, which determines the relative significance of graph self-expression (Eq. (2)), we set $\beta = [0.01, 0.1, 1, 5, 10, 20, 50, 100]$ and let two versions of JAT to classify nodes on the three smaller datasets. The corresponding results have been plotted in Fig. 3 (a) and (b). As the figures depict, both two versions of JAT may obtain a robust performance when different settings of $\beta$ are used. It is experimentally found that the variations of $\beta$ have a limited impact on the performance of JATs. Based on the obtained results, we set $\beta = 5$ for JATs when they perform the tasks of semi-supervised node classification on various datasets.

## D.4 PERFORMANCE ON NODE-WISE IMPLICIT DIRECTION STRATEGY

As defined in Eq. (6), $g_f$ and $g_s$ are layer-wise parameters which can capture the overall significance of graph structure and features in each layer of the neural architecture. They naturally can be node-level parameters in each layer of JAT, i.e., $g_{if}$ and $g_{is}$. As a result, the original *Implicit direction strategy* becomes *node-wise Implicit direction strategy* which can learn the relative significance of graph structure and features of each node when computing attention scores. And such *node-wise Implicit direction strategy* is defined as follows:

$$
r_{if} = \frac{\exp\left(g_{if}\right)}{\exp\left(g_{is}\right) + \exp\left(g_{if}\right)}, r_{is} = \frac{\exp\left(g_{is}\right)}{\exp\left(g_{is}\right) + \exp\left(g_{if}\right)},
$$
$$
\alpha_{ij} = \frac{r_{if} \cdot f_{ij} + r_{is} \cdot s_{ij}}{\sum_{k \in \mathcal{N}_i}\left[r_{if} \cdot f_{ik} + r_{is} \cdot s_{ik}\right]} = r_{if} \cdot f_{ij} + r_{is} \cdot s_{ij}.
$$

(39)

where $r_{if}$ or $r_{is}$ represents the normalized significance related to structure or node features associated with node $i$. Given the node-level joint attention mechanism defined in Eq. (39), JAT also can perform the task of representation learning in graph structured data. Here we compare its performance on transductive learning in datasets Cora, Citeseer, and Pubmed with the original two versions of JATs, i.e., JAT-I and JAT-E. The corresponding results are summarized in Table 5. As the table shows, the performance of JAT utilizing the attention mechanism of *node-wise Implicit direction strategy* is similar to that of JAT-E. According to the presented results, *node-wise Implicit direction strategy* is a potentially effective method through which JAT can learn meaningful representations for the subsequent predictive tasks. We will further investigate the effectiveness of this new strategy in our future works.

Table 5: Performance comparison on JATs using different attention strategies

| Different versions of JATs | Cora | Citeseer | Pubmed |
|---|---|---|---|
| JAT-E | 85.5±0.4% | 73.8±0.4% | 82.0±0.3% |
| JAT-I | 85.8±0.5% | 74.3±0.4% | 82.8±0.4% |
| JAT-I (node-wise) | 85.4 ±0.3% | 72.5±0.5% | 82.6±0.2% |

## E    DISCUSSIONS

In this section, we perform some detailed discussions that may help one to better understand the proposed JATs.

### E.1    MODEL COMPLEXITY

Based on the descriptions in Section 2, we may analyze the computational complexity of the proposed JATs. According to Eq. (3), for each node in the graph, the complexity for computing all the structural coefficients can be expressed as $O(N^2+N)$ for the worst case. However, as real graphs are always very sparse, the complexity for learning structural coefficients can be reduced to $O(e^2 + e)$, where $e$ represents average degree of the nodes. As the learning of each $\mathbf{C}_{i,:}$ is independent of others, Eq. (2) can be efficiently solved via parallel or distributed optimization strategies. For the computation of each attention head in the graph neural network, its complexity for learning $F^{l+1}$ features for each node can be represented as $O(ND^lD^{l+1}+(|E|+e)D^{l+1})$, which is same to the classical GAT. If there are $K$ attention heads used, the complexity becomes $O(K(ND^lD^{l+1} + (|E| + e)D^{l+1}))$.

### E.2    STRUCTURAL INFORMATION FOR COMPUTING ATTENTION SCORES

In the proposed JATs, structural coefficients learned from graph subspace are used for computing attention scores and the corresponding experimental performance is robust when compared with other state-of-the-art baselines. The module for learning structural coefficients is motivated by subspace clustering (Elhamifar & Vidal, 2013; Ji et al., 2014), which is an effective methodology for uncovering data clusters through performing the spectral analysis on the coefficient matrix ($\mathbf{C}$). Such matrix is also named as self-expressiveness matrix as it contains the coefficients which can construct a linear combination of each sample using others. Given Eq. (3), we know that the subspace learning module can directly infer the node-node correlations based on the graph data and those nodes sharing similar global structures (i.e., graph connectivity) may have higher structural coefficients. Rather than performing spectral analysis on $\mathbf{C}$, JAT directly makes use of the coefficients in it to compute attention scores for representation learning. This may allow JAT to pay more attention on those nodes which are structurally correlated when aggregating features passed to the higher layers.

Though an effective method for structural coefficients learning, the proposed module is only one of the alternatives that can be utilized by the generic framework JAT. As shown in the manuscript, $\mathbf{C}$ learned by the subspace learning module is different from the one learned by those methods for structural network embedding, such as Deepwalk (Perozzi et al., 2014), Node2Vec (Grover & Leskovec, 2016), Struc2vec (Ribeiro et al., 2017), and LINE (Tang et al., 2015). For such network embedding methods, the node-node correlation can be recovered via obtaining the inner product of node-wise representations. Given the fact that these network embedding methods also can effectively capture the node-node correlations, they are also effective alternatives for learning $\mathbf{C}$ for the proposed JAT. In the ablation study, we show that JAT may perform robustly even working with simple measures, e.g., Cosine and Jaccard similarity for obtaining node-node structural correlations, which indicates the proposed joint attention mechanisms are the main reasons boosting the performance of the attention-based GNNs. Thus, any measures/methods, that may effectively capture structural relationship between pairwise nodes in the graph, including those network embedding approaches and deep structural learning approach such as MoNet (Monti et al., 2017) can be used by JAT to capture node-node structural correlations for computing attention scores.

### E.3   RELATION BETWEEN JAT AND APPROACHES TO NETWORK EMBEDDING AND OTHERS

Concatenating the learned structural representations and corresponding node features, effective network embedding approaches, such as Deepwalk Perozzi et al. (2014), Node2Vec Grover & Leskovec (2016), Struc2vec (Ribeiro et al., 2017), and LINE (Tang et al., 2015) can also consider both structural and feature information for the subsequent learning tasks. Compared with them, the proposed JAT has its unique properties for learning graph representations. Rather than directly concatenating the structural and feature vectors, JAT utilizes novel joint attention mechanisms which may automatically determine whether graph structure or node feature is more important, to infer low-dimensional node representations via multi-layer feature aggregation. As $\mathbf{C}$ is used for computing attention scores, the node representations learned by JAT are automatically implanted with meaningful structural information hidden in the graph. It is seen that the strategy for automatically determining the relative significance between graph structure and vertex features and the verified expressive power distinguish the proposed JAT from other related works. Besides, for attention-based GNNs, the proposed joint attention mechanisms are also novel attempts to effectively consider structural information for computing attention scores. No previous mechanisms for GNNs can consider learning the relative significance between graph structure and node features for computing attentions scores. The proposed JAT is also different from those GNNs considering structural information, e.g., diffusion (Klicpera et al., 2019; Busch et al., 2020) or GCNs for subspace clustering (Cai et al., 2020). Those diffusion based GNNs mainly consider higher-order structural information among node pairs to formulate the functions for feature aggregation. As for GNNs for subspace clustering, their main task is to learn the self-expressive matrix, i.e., $\mathbf{C}$, based on the deep neural networks and then data clusters are inferred via performing spectral analysis on the learned $\mathbf{C}$. Compared with them, JAT directly makes use of $\mathbf{C}$ and node features to compute attention scores for aggregating the features passed to the higher layers. $\mathbf{C}$ in the proposed JAT plays an important role in obtaining appropriate feature aggregations, rather than clustering. Given the mentioned facts, the proposed JAT is novel compared with existing methods.

### E.4   LIMITATIONS OF JAT

Though effective in predicting node labels in transductive learning tasks, the proposed JAT has some limitations to overcome in the future. First, the efficiency for learning $\mathbf{C}$ in the current version of JAT is somewhat unsatisfying. Based on the analysis on computational complexity, JAT has to pre-train $\mathbf{C}$ for those massive datasets (e.g., OGBN-Arxiv) due to the limited computing resources. More efficient techniques for capturing the node-node structural correlations can therefore be considered by JAT. Based on the results shown in the ablation studies, the proposed joint attention mechanisms ensure JAT to perform robustly as long as the used node-node correlations can effectively reveal the structural properties of the graph. Thus, those network embedding approaches and other efficient methods for deep structural learning can be combined with the generic JAT. Second, the expressive power of JATs is theoretically validated mainly in the countable feature space in this paper. In those graphs whose features are in uncountable space, the expressive power of JATs needs re-verifying following the recent framework proposed in (Corso et al., 2020). As mentioned in (Corso et al., 2020), some theorems related to the expressive power in the countable feature space might not always hold in the feature space uncountable. To the theoretical perspective, attention mechanisms which are analogously powerful in both countable and uncountable feature spaces are thereby needed by both JATs and other attention-based GNNs.

