# OpenReview forum: "Graph Joint Attention Networks"
_ICLR.cc/2021/Conference — Reject_

### Official Review · AnonReviewer3 · 2020-10-26
**Interesting idea, limited experiments**

**Rating:** 5
**Confidence:** 3

**Review:**

---- Summary

This paper proposes Graph Joint Attention Networks (JAT), which augment Graph Attention Networks (GAT) by introducing structural attention coefficients, which are combined with the feature based attention coefficients computed by GAT. The authors present two ways of incorporating the structural coefficients, namely Implicit direction and Explicit direction. The authors evaluate their proposed technique in the Cora, Citeseer and Pubmed datasets, where they improve the performance of other GNN approaches.

---- Pros
* The paper presents an interesting approach to introduce structural information when computing the attention coefficients in an attention-based GNN.
* Good analysis of the expressive power of JAT
* The method presented in the paper achieves state of the art results in the datasets it is evaluated with.

--- Cons
* The empirical evaluation is a bit limited, since it evaluates only with Cora, Citesser and Pubmed which have been shown to be not entirely appropriate to evaluate GNNs, in favour of better benchmarks like the ones proposed in [1] and [2].
* It isn’t very clear exactly how the minimization of Equation 2 is done. Is it solved once for each graph or is it solved at every training iteration? The authors say that “JATs can optimize Eq. (2) together with the training of the neural architecture.” so it isn’t clear how it is done. If A is always the same for each dataset (Cora, Citeseer and Pubmed are one single graph each) why not minimize it just once before training?
* The authors don’t mention the complexity of their approach. If JAT were to be applied to a different graph structure all the time, so A is different for each sample in a dataset consisting of multiple graphs, what would be the complexity of the method considering it has to minimize Equation 2 to obtain the matrix of coefficients C?
* The experimental section lacks some ablation studies. For example, what if only the structural attention coefficients are used? What happens if noise is included in the adjacency matrix? What if structural coefficients are all constant? Then the model should be empirically equivalent to GAT.

---- Questions
* For very different values of $\beta$ (ranging from 0.01 to 100), JAT seems to achieve very similar performance, a behaviour consistent across datasets. Any idea why that is the case?
* The implicit direction mechanism is a bit unclear in the paper. What are $r_s$ and $r_f$ in Eq 6? They have not been introduced before and it isn’t clear where do the indexes $f$ and $s$ come from. I think Eq 5 would be a bit clearer if written like $$ r_i = \frac{exp(g_i)} {\sum_{k \in \\{s,f\\}} exp(g_k)}, i \in \\{s, f\\} $$ Although it is still a bit confusing, using $i$ as an index when it has been previously used to denote node indexes is confusing.
* $g_s$ and $g_f$ are global parameters, meaning that each node structural and feature coefficients are weighted in the same way. Why not learn these parameters for each node, so that some nodes can rely more on features and other nodes can rely more on structure?
* Could JAT work in the inductive case where the training and testing structures are different?

---- Minor Comments

The font-size in the plots in Figure 2 should be increased, it would be very difficult to read if the paper was printed.

---- Reason for score

The paper proposes an interesting technique to use the graph structure as well as the nodes features to compute attention coefficients, however the experiments are a bit limited, using only three small datasets and limited ablation studies. Additionally, some of the details of the paper are not clear. For these reasons, I consider the paper to be marginally below the acceptance threshold.

------------ Post rebuttal update

I appreciate the rebuttal made by the authors, as they have clarified my questions about the paper. It is also good that a new dataset and additional ablation experiments have been added to the paper, however, the empirical section still needs to be improved in my opinion.
For example, more datasets and stronger baselines could be used, and I believe the authors could do it if they had more time. Therefore, even though I mantain my score, I encourage the authors to strengthen the empirical evaluation for future submissions.

---- References

[1] Dwivedi, V. P., Joshi, C. K., Laurent, T., Bengio, Y., & Bresson, X. (2020). Benchmarking graph neural networks. arXiv preprint arXiv:2003.00982.

[2] Hu, W., Fey, M., Zitnik, M., Dong, Y., Ren, H., Liu, B., ... & Leskovec, J. (2020). Open graph benchmark: Datasets for machine learning on graphs. arXiv preprint arXiv:2005.00687.

---

> ### Author Response · Authors · 2020-11-24
> **Author response to Reviewer 3**
>
> The authors thank the Reviewer for raising constructive comments for improving the paper.
>
> Q1: The empirical evaluation is a bit limited, since it evaluates only with Cora, Citesser and Pubmed which have been shown to be not entirely appropriate to evaluate GNNs, in favour of better benchmarks like the ones proposed in [1] and [2].
>
> Response to Q1: Thank you very much for raising such an invaluable suggestion that can further validate the effectiveness of different GNNs. As suggested, in the revised manuscript, we first briefly mention that those three citation networks, i.e., Cora, Citesser and Pubmed are insufficient for testing the effectiveness of different GNNs, following the statement made in [1] and [2]. Then, we perform additional experiments on dataset OGBN-Arxiv, which is released in [2]. And the corresponding experimental results have presented in Table 2 in the revised manuscript. As the table shows, the proposed JATs still can achieve the state-of-the-art performance when the new dataset is used.
>
> Q2: It isn’t very clear exactly how the minimization of Equation 2 is done. Is it solved once for each graph or is it solved at every training iteration? The authors say that “JATs can optimize Eq. (2) together with the training of the neural architecture.” so it isn’t clear how it is done. If A is always the same for each dataset (Cora, Citeseer and Pubmed are one single graph each) why not minimize it just once before training?
>
> Response to Q2: The mentioned issue has been clarified in the revised manuscript (See Appendix). Specifically, we point out in the revised manuscript that, the minimization of Eq. 2 can either be solved at every training iteration, or before training the graph neural architecture and C has to be obtained in different graphs (e.g., the mentioned Cora, Citeseer and Pubmed) as their structures are different. If Eq. 2 is minimized iteratively together with the training of JAT, the optimal structural coefficients in C can be obtained when the training of JAT is also convergent. Thus, structural coefficients may gradually influence the representation learning.  Eq. 2 can also be optimized prior to the training of JAT. Then JAT will concentrate on learning embeddings, layer-wise attention weights, and relative significance between structural and feature attentions. It should be noted that minimizing Eq. 2 before training JAT may significantly improve the efficiency of JAT when it is used in larger datasets, e.g., Ogbn-Arxiv, while its performance is still robust. Such an observation also indicates the effectiveness of the proposed attention mechanisms adopted by JAT.
>
> Q3: The authors don’t mention the complexity of their approach. If JAT were to be applied to a different graph structure all the time, so A is different for each sample in a dataset consisting of multiple graphs, what would be the complexity of the method considering it has to minimize Equation 2 to obtain the matrix of coefficients C?
>
> Response to Q3: The mentioned issue has been addressed in the revised manuscript. We analyze the computational complexity of the proposed model. Specifically, we point out in the revised manuscript that (Appendix E. 1), the complexity for learning C for each node is O(e^2 + e), where e represent the average node degree in the graph. For each attention head in JAT, the complexity for obtaining D^{l+1} features for each node is $O(ND^lD^{l+1}+(\vert E\vert + e) D^{l+1})$, which is same to GAT. The complexity of K heads attention is $O(K(ND^lD^{l+1}+(\vert E\vert + e) D^{l+1}))$.
>
> Q4: The experimental section lacks some ablation studies. For example, what if only the structural attention coefficients are used? What happens if noise is included in the adjacency matrix? What if structural coefficients are all constant? Then the model should be empirically equivalent to GAT.
>
> Response to Q4: As suggested, more ablation studies are performed, and the corresponding are presented in the revised manuscript. Specifically, we make comprehensive comparisons on different versions of JATs using different information to compute attention scores and the corresponding performance on node classification is reported in Table 3 in the revised manuscript. As the table shows, when JAT uses some corrupted information, e.g., only structural coefficients, its performance is slightly worse than the original version JAT.

---

> > ### Author Response · Authors · 2020-11-24
> > **Author response to Reviewer 3 (Cont.)**
> >
> > Q5: For very different values of (ranging from 0.01 to 100), JAT seems to achieve very similar performance, a behaviour consistent across datasets. Any idea why that is the case?
> >
> > Response to Q5: The reason that leads to the mentioned observation can be revealed if we take the partial derivative of Eq. (2). As l_1 norm is used for sparsity control in the learning of C, the gradient of l_1 norm is a constant, which is much smaller than the gradient related to the regularization term. Thus, the regularization term in Eq. (2) dominates the learning of C in our problem and Cs obtained using different \beta are similar to some extent. The performance on node classification is not heavily influenced by the variations of \beta.
> >
> > Q6: The implicit direction mechanism is a bit unclear in the paper. What are  and  in Eq 6? They have not been introduced before and it isn’t clear where do the indexes  and  come from. I think Eq 5 would be a bit clearer if written like
> > Although it is still a bit confusing, using  as an index when it has been previously used to denote node indexes is confusing.
> >
> > Response to Q6: As suggested, Eq. (5) (Eq. (6) in the revised manuscript) has been rewritten to avoid any confusion in the revised manuscript.
> >
> > Q7: and  are global parameters, meaning that each node structural and feature coefficients are weighted in the same way. Why not learn these parameters for each node, so that some nodes can rely more on features and other nodes can rely more on structure?
> >
> > Response to Q7: Thanks very much for the suggestion. In the revised manuscript, we point out that g_s and g_f are layer-wise parameters (not global) that may capture the overall structural and feature preference in each layer when computing the joint attention scores. Certainly, it can be set as node-wise parameters, which brings more computing units to the neural architecture. For layer-wise setting, computing g_s/g_f needs \vert E \vert units of gradient, while it becomes eN units of gradient if they are set as node-wise parameters. We will attempt to investigate the effectiveness of such node-wise setting in our future works. Additionally, we perform some experiments comparing the node-level implicit direction strategy with original versions of JATs. The results can be checked in the appendix. We will further explore its effectiveness in our future works.
> >
> > Q8: Could JAT work in the inductive case where the training and testing structures are different?
> >
> > Response to Q8: Theoretically, JAT can work in the inductive case where the training and testing structures are different. As a message-passing GNN for learning representations in graphs, JAT can learn the node-level representations through multi-layer aggregation of node features. These learned representations can be used in inductive learning tasks by further performing an effective readout operation (readout function). As the main contribution of this paper is to propose novel joint attention mechanisms to guide the attention-based GNNs to pay more attention on those neighbors that are structurally and feature correlated, we mainly use transductive learning tasks to test the effectiveness of JAT. We will further test the effectiveness of JAT in inductive learning in our future works.
> >
> > Q9: The font-size in the plots in Figure 2 should be increased, it would be very difficult to read if the paper was printed.
> >
> > Response to Q9: As suggested, Fig. 2 has been redrawn to make it clear to read.
> >
> > Q10: The paper proposes an interesting technique to use the graph structure as well as the nodes features to compute attention coefficients, however the experiments are a bit limited, using only three small datasets and limited ablation studies. Additionally, some of the details of the paper are not clear. For these reasons, I consider the paper to be marginally below the acceptance threshold.
> >
> > Response to Q10: As suggested, in the revised manuscript, we briefly discuss the problems when the mentioned three classical networks are used by citing the related works. We additionally include OGBN-Arxiv, which is available in Open graph benchmark database to test the effectiveness of the proposed model. Then, more ablation studies, e.g., allowing JATs to adopt different strategies to learn structural correlations to perform graph learning tasks are performed in the revised manuscript. The corresponding results can be seen in Section IV and Appendix D in the revised manuscript. At last, all the mentioned confusing issues are clarified in the revised manuscript.

---

### Official Review · AnonReviewer4 · 2020-10-28
**An interesting and good paper**

**Rating:** 7
**Confidence:** 4

**Review:**

Overall Comments:
Creating aggregation weight over neighbor nodes lies at the key part of graph neural networks. Generally the weights can be generated by the node structure or feature similarity. The node structure similarity provides a way to measure the correlation of a pair of nodes with a complete graph. While attention weight usually focuses on the local neighborhood, which can be easily biased by the node popularity [1, 2]. If a neighbor node with large degree, it potentially tends to have a large embedding norm. However, from complete view of graph structure, this neighbor might be a noisy node. With a given complete graph structure, we can downgrade the influence of structure bias. From this point of view, it's interesting to study whether the attention and structure weights can be complementary to each other. The key idea of this work is very easy to follow. The proposed strategy is to unify the weights generated by graph structure and node feature. Theoretical analysis shows the expressive power of  the proposed strategy. The experimental results for node classification also demonstrates its superiority comparing with state-of-the-art baselines.

Presentation:
The presentation of this work is very clear and organized very well.

Questions:
1. The coefficient matrix C is learned by recovering the adjacent matrix A. To some extent, it tells the correlation between a pair of nodes. Can we replace it with another correlation matrix, such as Jaccard similarity matrix where each element stands for the node similarity measured by Jaccard equation or Personalized PageRank diffusion matrix? That'd be interesting to see the influence coming from the coefficient matrix obtained by different methods.

2. In Section 2.1, what is the role of the multiset M_i? It's a little difficult to understand without any background introduction in this subsection. Maybe more justification about the notations are needed to show its connection to next section.

References:

[1] Gong, Chengyue, Di He, Xu Tan, Tao Qin, Liwei Wang, and Tie-Yan Liu. Frage: Frequency-agnostic word representation. In Advances in neural information processing systems, pp. 1334-1345. 2018.

[2] Armandpour, Mohammadreza, Patrick Ding, Jianhua Huang, and Xia Hu. "Robust negative sampling for network embedding." In Proceedings of the AAAI Conference on Artificial Intelligence, vol. 33, pp. 3191-3198. 2019.

---

> ### Author Response · Authors · 2020-11-24
> **Author response to Reviewer 4**
>
> The authors thank the Reviewer to raise constructive comments that may improve the paper.
>
> Q1: The coefficient matrix C is learned by recovering the adjacent matrix A. To some extent, it tells the correlation between a pair of nodes. Can we replace it with another correlation matrix, such as Jaccard similarity matrix where each element stands for the node similarity measured by Jaccard equation or Personalized PageRank diffusion matrix? That'd be interesting to see the influence coming from the coefficient matrix obtained by different methods.
>
> Response to Q1: Thanks for raising this suggestion that may further investigate the effectiveness of JATs. As described in Section 2.4, many effective approaches/metrics that can obtain the structural coefficients, such as the mentioned Jaccard similarity and Personalized PageRank diffusion, can be adopted by JATs to learn graph representations. In Ablation study of the revised manuscript, we use JATs that integrate different strategies to learn node-node structural coefficients to perform semi-supervised node classification in all the testing datasets. As the results show, JATs still perform robustly when different strategies for obtaining structural coefficients are used. This indicates that the proposed joint attention mechanisms are very effective in learning meaningful graph representations for the subsequent learning/analytical tasks.
>
> Q2: In Section 2.1, what is the role of the multiset M_i? It's a little difficult to understand without any background introduction in this subsection. Maybe more justification about the notations are needed to show its connection to next section.
>
> Response to Q2: As suggested, the role of the multiset M_i has been clarified in the revised manuscript to make the readers better understand the concept. Specifically, in Section 2.1 of the revised manuscript, we explain that given the features of a central node (say node i) and its neighbors, they form a multiset M_i = (S_i, mu_i), where S_i is the underlying set containing the distinct elements of M_i, and mu_i is the multiplicity of each element in S_i. Please see Section 2.1 for more details.

---

### Official Review · AnonReviewer1 · 2020-10-28
**Improved expressivity on structural GATs, lacking experimental protocol**

**Rating:** 5
**Confidence:** 4

**Review:**

The authors propose Graph Joint Attention Networks (JATs) as a mechanism for explicitly incorporating structural information in the GATs attention mechanism.

JATs separately learn a "structural" attention matrix which:

(a) exploits structural correlations between vertices;
(b) forces diagonal entries to be zero to avoid trivial solutions (clever!);
(c) is regularised using the L_1 norm, to promote sparsity.

Such a matrix is then recombined with the attention matrix computed by vanilla GATs, yielding updated attentional coefficients which empirically improve on GATs on several benchmark datasets, and can be proven (with small modifications) to satisfy the equivalence with the 1-WL test, which also answers a somewhat-open question on GATs' expressive power.

Overall, I found the work's theoretical contribution to be impressive. However I cannot recommend acceptance in the current form based on the theory alone, and in particular, the following three major points should be addressed by the authors before the paper is ready:

- The authors only evaluate their work on the Cora/Citeseer/Pubmed citation networks, which are known to be oversaturated and unreliable as graph representation learning benchmarks (see e.g. "Pitfalls of Graph Neural Network Evaluation" from Shchur et al.) and should be avoided as centerpieces of evaluation. I would like to see experimental results on either OGB (Hu et al., NeurIPS'20) or Benchmarking-GNNs (Dwivedi et al.), which are two established datasets that have been around for long enough by now, and are easily accessible via existing graph learning libraries.
- The proposed structural learning work is one way of injecting structural information into the attentive mechanism, but is not the only one. Actually, taking either:
top/bottom-k eigenvectors of the Graph Laplacian, or
a structural embedding method such as DeepWalk/node2vec/struct2vec/LINE,
as additional structural features to be concatenated with node features, could yield a similar effect of exploiting structural similarities without necessitating the need for two attentional mechanisms or additional learning objectives. Also it might be interesting to contrast the method proposed here with MoNet (already cited), which only performs attention on structural features -- maybe MoNet coefficients could serve as a possible replacement for the C matrix. Could the authors compare with some of the above, either theoretically or empirically?
- Lastly, and in relation to the previous point, I found the following claim by the authors:
"First, there are no appropriate attention mechanisms which can automatically identify the relative significance between the latent structure and node features while computing the attention scores."
to be very strong and unlikely to be correct, especially in light of the discussion mentioned above. Also the phrase "latent structure" is only used there and not properly defined elsewhere, so it is unclear what the authors are referring to.

I am willing to increase my score if the above were appropriately addressed. Besides this, there are a few other points that could also be quite useful to consider:

- The paper follows the theoretical assumptions of the GIN paper (countable feature-spaces), and it could be interesting to also look at uncountable feature-spaces (eg real numbered features) where many of the theoretical results don't hold. See Corso, Cavalleri et al. (NeurIPS'20) "Principal Neighbourhood Aggregation for Graph Nets" for an overview.
- I'm a bit concerned about the method's scalability. It seems that solving the optimisation task on C may easily invoke quadratic memory even if the underlying graph is sparse? Have the authors managed to reduce this constraint somehow, given that their method successfully ran on Pubmed? Further, some proposed approaches above (e.g. DeepWalk features or using MoNet attention to compute C) could be more scalable alternatives. Could the authors comment on this?
- Finally, isomorphism-style setups are concerned with distinguishing entire (sub)graphs, and hence might be more directly amenable to graph classification style tasks. The authors may wish to consider some classification tasks (maybe even synthetically constructed) in order to solidify the predictive power argument of their method.

==================== Post-rebuttal update:

I would like to thank the authors for providing a detailed reply, which partially addresses some of my concerns. However, there are several issues in the reply and updated manuscript as provided by the authors:

- The results on OGBN-arXiv are definitely useful, but there is no GAT-based baseline, which corresponds to the JAT's main comparison point. Therefore it is not possible to make strong claims about the method's effectiveness based on the following baseline choice alone.
- The discussion of alternate structural learning approaches is not sufficiently detailed, and has no empirical backing to the authors' proposal. More experimentation is needed, in my opinion, to fully back the authors' claim of: "However, these methods may not appropriately determine which part, i.e., structure or node features is more important in the embedding space."
- Lastly, I do not find that the authors have appropriately toned down their claims in the Introduction. Most critically, the authors write: "The experimental results show that JATs achieve the state-of-the-art performance". A lookup of the OGBN-arXiv publicly available leaderboard demonstrates that this is not the case: https://ogb.stanford.edu/docs/leader_nodeprop/#ogbn-arxiv

Having state-of-the-art is not the most important thing, but it should not be claimed when it's not achieved.

Overall I think the JAT paper has a lot of potential. For now I choose to retain my 'weak reject' recommendation, and hope the authors will take my comments into account for subsequent submissions.

---

> ### Author Response · Authors · 2020-11-24
> **Author response to Reviewer 1 Part 1**
>
> The authors thank the Reviewer for raising constructive comments to improve the paper.
>
> Q1: The authors only evaluate their work on the Cora/Citeseer/Pubmed citation networks, which are known to be oversaturated and unreliable as graph representation learning benchmarks (see e.g. "Pitfalls of Graph Neural Network Evaluation" from Shchur et al.) and should be avoided as centerpieces of evaluation. I would like to see experimental results on either OGB (Hu et al., NeurIPS'20) or Benchmarking-GNNs (Dwivedi et al.), which are two established datasets that have been around for long enough by now, and are easily accessible via existing graph learning libraries.
>
> Response: Thanks very much for raising this invaluable suggestion that may further investigate the effectiveness of the proposed JAT. Following the suggestion, we have tested JAT using dataset OGBN-Arxiv, which is available in Open graph benchmark database, and compared the performance of JAT with other baselines.
>
> Specifically, by citing the mentioned works, we point out in the revised manuscript that, only using the three classical citation networks to test the performance of different approaches is insufficient, we also use OGBN-Arxiv to test the effectiveness of different approaches and the corresponding results are presented in Table 1 and 2. As depicted, the proposed JAT achieves best performance in various testing datasets, which indicates the effectiveness of the proposed GNN.

---

> > ### Author Response · Authors · 2020-11-24
> > **Auther response to Reviewer 1 Part 2**
> >
> > Q2: The proposed structural learning work is one way of injecting structural information into the attentive mechanism, but is not the only one. Actually, taking either: top/bottom-k eigenvectors of the Graph Laplacian, or a structural embedding method such as DeepWalk/node2vec/struct2vec/LINE, as additional structural features to be concatenated with node features, could yield a similar effect of exploiting structural similarities without necessitating the need for two attentional mechanisms or additional learning objectives. Also it might be interesting to contrast the method proposed here with MoNet (already cited), which only performs attention on structural features -- maybe MoNet coefficients could serve as a possible replacement for the C matrix. Could the authors compare with some of the above, either theoretically or empirically?
> >
> > Response to Q2: Thanks very much for raising this suggestion that may further improve the utility of JAT. Indeed, there are many other methods that can be used to inject structural information into node features and the concatenated representations can be used for subsequent prediction tasks. However, these methods may not appropriately determine which part, i.e., structure or node features is more important in the embedding space. For example, those mentioned methods for network embedding attempt to learn node embeddings that are composed of two representations from graph topology and node feature. But there are always no effective strategies for automatically determining the relative significance between them. This is one of reasons motivate us to propose JAT.
> >
> > To better distinguish the proposed approach from other related ones, we perform a detailed discussion in Appendix E in the revised manuscript. As mentioned in the manuscript, the main contribution of this manuscript is to propose novel attention-based GNNs that utilize novel joint attention mechanisms to effectively compute the attention scores considering both structural and feature information carried by the graph. This means JAT can take into consideration both structural and feature information and automatically pay more attention to those node pairs which are structurally and contextually correlated. The proposed attention mechanisms make JAT different from those mentioned methods for network embedding as they allow JAT to automatically determine whether graph structure or node feature is more important for representation learning. Besides, the expressive power of the attention-based function for feature aggregation meets the 1-WL test. This means JAT can combine with any effective methods for computing/learning structural information hidden in the graph to compute attention scores for learning representations in the graph. As the learning outputs of all the network embedding methods and MoNet mentioned by the reviewer can directly or indirectly represent node-node correlations, they are all possible alternatives for replacing the subspace learning module to learn C.
> >
> > To further explore the utility of JAT, in the revised manuscript, we briefly discuss whether the propose GNN can use the methods mentioned by the reviewer to compute the attention scores for feature aggregation. We point out in the revised manuscript that, according to the joint attention mechanisms proposed and the presented theoretical analysis on the expressive power in the manuscript, any effective methods for computing/learning structural correlations/similarities, including those mentioned by the reviewer can be used by JAT to compute attention scores and the layer-wise attention-based function for feature aggregation is same as what 1-WL test does when distinguishing different graph structures.
> >
> > At last, we let JAT combine with different strategies (such as Cosine, and Jaccard similarity) for obtaining node-node structural correlations in the graph to compute attentions scores for representation learning and test the corresponding performance. The experimental results have been reported in Ablation study in the revised manuscript. As the results show, JAT still performs robustly when combining with strategies other than the subspace learning, which indicates the effectiveness of the proposed joint attention mechanisms. Given such observations, it is said that all those methods mentioned by the reviewer can be used by the generic joint attention mechanisms proposed in the manuscript.

---

> > > ### Author Response · Authors · 2020-11-24
> > > **Author response to Reviewer 1 Part 3**
> > >
> > > Q3: Lastly, and in relation to the previous point, I found the following claim by the authors: "First, there are no appropriate attention mechanisms which can automatically identify the relative significance between the latent structure and node features while computing the attention scores." to be very strong and unlikely to be correct, especially in light of the discussion mentioned above. Also the phrase "latent structure" is only used there and not properly defined elsewhere, so it is unclear what the authors are referring to.
> > >
> > > Response to Q3: The mentioned issue has been addressed to avoid any confusion in the revised manuscript. Specifically, we point out in the revised manuscript that: For attention-based GNNs, appropriate attention mechanisms which can automatically identify the relative significance between the graph structure and node features while computing the attention scores are not many. Besides, we use "graph structure" to replace "latent structure" in the revised manuscript.
> > >
> > > Q4 : The paper follows the theoretical assumptions of the GIN paper (countable feature-spaces), and it could be interesting to also look at uncountable feature-spaces (eg real numbered features) where many of the theoretical results don't hold. See Corso, Cavalleri et al. (NeurIPS'20) "Principal Neighbourhood Aggregation for Graph Nets" for an overview.
> > >
> > > Response to Q4: As suggested, we briefly discuss the issue mentioned by the reviewer in Appendix E and we also state that the validations in this paper assume the feature space is countable in the revised manuscript. Specifically, when facing the uncountable feature spaces as mentioned in the NeurIPS'20 paper, the proposed joint attention mechanisms might fail to distinguish some (sub)-graphs, although their expressive power has been verified in the countable space. Intuitively, JAT cannot distinguish different (sub-)graphs in the uncountable space whose expectations of node features plus degree-related central node features are the same. This is one of the limitations of the proposed GNN. We will attempt to propose novel joint attention mechanisms that may address the limitation.
> > >
> > > Q5: I'm a bit concerned about the method's scalability. It seems that solving the optimisation task on C may easily invoke quadratic memory even if the underlying graph is sparse? Have the authors managed to reduce this constraint somehow, given that their method successfully ran on Pubmed? Further, some proposed approaches above (e.g. DeepWalk features or using MoNet attention to compute C) could be more scalable alternatives. Could the authors comment on this?
> > >
> > > Response to Q5: As mentioned by the reviewer, the scalability related to the learning of C is an issue for the proposed JAT. In the revised manuscript, we analyze the computational complexity of the proposed model, including the learning of C. We also provide two training strategies for obtaining C:
> > > 1. Learning C along with the training of the neural network. This means we formulate a loss function summing the cross-entropy error for representation learning and the subspace learning.
> > > 2. Pretraining C before training JAT. This may significantly improve the efficiency of JAT when it deals with large scale data, e.g., OGBN-Arxiv.
> > >
> > > Just as the discussions to other questions, the learning method for C can be anyone that is effective in capturing the structural correlations between pairwise nodes. Thus, it is certain that the subspace learning of C can be replaced by other methods, such as DeepWalk and MoNet. In the revised manuscript, we let JAT make use of different strategies to learn structural features to compute attention scores for graph representation learning and test the corresponding performance. The experimental results have been reported in Table 3 in the revised manuscript.
> > >
> > > Q6: Finally, isomorphism-style setups are concerned with distinguishing entire (sub)graphs, and hence might be more directly amenable to graph classification style tasks. The authors may wish to consider some classification tasks (maybe even synthetically constructed) in order to solidify the predictive power argument of their method.
> > >
> > > Response to Q6: As suggested, the proposed JAT has been tested on more datasets, i.e., OGBN-Arxiv. Although it is still a dataset related to node classification tasks, its size and structure are challenging for many GNN models. Thus, it is a good testbed for the predictive power of graph learning approaches. We have presented the corresponding results in Section IV in the revised manuscript. As the results show, the proposed JAT still can achieve state-of-the-art performance when more datasets are used for testing.
> > >
> > > In future, we will further test the effectiveness of JAT in graph classification tasks by combing JAT with effective readout functions which either exist or are proposed.

---

### Official Review · AnonReviewer2 · 2020-10-28
**Introduce subspace clustering information into a graph-attention network to capture non-local information into the node embeddings**

**Rating:** 4
**Confidence:** 5

**Review:**

The paper discusses how to introduce some form on non-local structural information into the node representations by collating cluster similarity information from a topological subspace clustering with a classical spatial graph convolution. Both information are mediated by attentional mechanisms which also take care of mixing the topological and convolutional contributions.

On the positive side, the proposed approach seems to work well empirically, although results are limited to 3 standard benchmarks on node classifications. The community is trying to surpass the use of those 3 datasets, due to both their simplistic nature (see ogb.stanford.edu for challenging node classification benchmarks) and pitfalls in their evaluation (https://arxiv.org/pdf/1811.05868.pdf).

The paper also provides a theoretical proof of the representation power of the proposed JAT, following the consolidated approach started in the GIN paper. I am not particularly fond of these proofs as they are mostly technically correct but poorly original exercises showing that the proposed method is more or less 1-WL equivalent. In the paper, such proofs are leveraged to show a class of substructures on which the vanilla JAT model fails, providing a correction that solves the issue. Which would be fine, apart from the fact that the proposed work-around (i.e. considering neighbourhood size of each node) is standard practice in node classification tasks, where node degree is added as a node feature in input to the model (with excellent accuracy boosts).

This brings me to the big issue in the paper: originality is minor. The work is heavily based on the GAT model extended, again by a fairly standard attention-based approach, with the topological clustering information. This latter aspect is allegedly the novel part of the work, but the paper does not analyse the implications of adding such topological information in full depth. The paper is very generic in referring to it as an enabler for considering higher order structural information, without providing any insight onto the nature of such added information. To me, the introduction of subspace clustering information is somehow pushing some spectral information into the spatial convolution-based approach by GAT. In this sense, I see the proposed JAT as an attempt to trade between spatial and spectral features. In the literature there are works providing evidence on how mixing the efficient, but local, approach of spatial convolutions with the global information from spectral methods typically yields to increased predictive accuracies (see papers.nips.cc/paper/9490-diffusion-improves-graph-learning.pdf). I would have liked to read in the paper a discussion placing the proposed approach in this recent interesting literature, e.g. leveraging diffusion information (as in the paper above or https://ecai2020.eu/papers/1339_paper.pdf) or very related subspace clustering methods (https://ieeexplore.ieee.org/document/9181470).  Instead, the discussion of GNN background (Section 1) and related methods (Section 2.) is quite shallow. This is a pity (and the second major issue in the paper) and it reduces considerably the impact of the work, especially considering the scope of the ICLR conference.

There is a third issue which relates to some limitations of the approach which are more or less obfuscated in the paper. The work focuses on node classification tasks, ie. where data comprises a single network. Why is this? Is it because the method cannot handle a dataset of graphs of varying topology (e.g. as in graph classification)? Similarly, why (in Section 2.1) the model is restricted to work with only binary node features? Is this limitation purely notational or does it affect the model? The manuscript should be very clear in stating the limitations of the proposed approach, if any, and what bit of the model is possibly introducing such constraints.

In the empirical analysis, Figure 2.c should also report the confidence intervals, as I am not convinced that the difference “with and without ad” is statistically significant.

The denominator in (5) can be simplified as it is a summation over two elements.

=========== POST REBUTTAL

I have really appreciated the effort placed by the Authors in their rebuttal. Here follows some considerations following rebuttal:
* The empirical analysis has been strenghtened with the inclusion of the OGB benchmark as it provides a view of the empirical performance of JAT in the context of a challenging dataset within a standardised setup. At the same time, OGB allows direct comparison with leaderboards and results in literature that highlight how JAT performance are not (as claimed) state-of-the-art. In this sense, it is still quite unclear if there is any practical advantage with respect to GAT.
* The revised paper version places the work much better in the context of recent related literature.
* Also considering the point above, the novelty of the work is still borderline to me.

In summary, the work has good potential, but it is not yet ready, hence my decision to stay on the reject side. The contained originality can be made up in a future submission by providing convincing state-of-the-art performance results on challenging benchmarks, showing substantial differences from related models such as GAT.

---

> ### Author Response · Authors · 2020-11-24
> **Author response to Reviewer 2 Part 1**
>
> The authors thank the Reviewer for the constructive comments and suggestion which may help to further improve the paper.
>
> Q1: On the positive side, the proposed approach seems to work well empirically, although results are limited to 3 standard benchmarks on node classifications. The community is trying to surpass the use of those 3 datasets, due to both their simplistic nature (see ogb.stanford.edu for challenging node classification benchmarks) and pitfalls in their evaluation (https://arxiv.org/pdf/1811.05868.pdf).
>
> Response to Q1: The mentioned issue has been addressed in the revised manuscript. More datasets have been used to test the effectiveness of different approaches. Specifically, we use OGBN-Arxiv to test the effectiveness of different approaches and the corresponding results are presented in Table 2. As depicted, JAT still can achieve the state-of-the-art performance by using a relatively simple network structure (2-layer network structure, please see the details in Appendix), which indicates the robustness of the proposed GNN. The augmented experimental results may validate the effectiveness of the proposed JAT. Besides, we briefly discuss why more datasets should be included into the experiments by citing the mentioned article.

---

> > ### Author Response · Authors · 2020-11-24
> > **Author response to Reviewer 2 Part 2**
> >
> > Q2: This brings me to the big issue in the paper: originality is minor. The work is heavily based on the GAT model extended, again by a fairly standard attention-based approach, with the topological clustering information. This latter aspect is allegedly the novel part of the work, but the paper does not analyse the implications of adding such topological information in full depth. The paper is very generic in referring to it as an enabler for considering higher order structural information, without providing any insight onto the nature of such added information. To me, the introduction of subspace clustering information is somehow pushing some spectral information into the spatial convolution-based approach by GAT. In this sense, I see the proposed JAT as an attempt to trade between spatial and spectral features. In the literature there are works providing evidence on how mixing the efficient, but local, approach of spatial convolutions with the global information from spectral methods typically yields to increased predictive accuracies (see papers.nips.cc/paper/9490-diffusion-improves-graph-learning.pdf). I would have liked to read in the paper a discussion placing the proposed approach in this recent interesting literature, e.g. leveraging diffusion information (as in the paper above or https://ecai2020.eu/papers/1339_paper.pdf) or very related subspace clustering methods (https://ieeexplore.ieee.org/document/9181470). Instead, the discussion of GNN background (Section 1) and related methods (Section 2.) is quite shallow. This is a pity (and the second major issue in the paper) and it reduces considerably the impact of the work, especially considering the scope of the ICLR conference.
> >
> > Response to Q2: Thanks very much for raising this suggestion which may significantly improve the paper. The mentioned issue has been addressed in the revised manuscript. First, we further analyze why subspace learning may effectively capture the structural correlations (Please see Eq. (2) and (3), and corresponding discussions.) Second, we point out in the revised manuscript that JAT indeed attempts to leverage more structural information (but not necessarily spectral information) than empirical attention-based GNNs, e.g., GAT to learn meaningful representations in graph data. We propose two novel, and generic attention mechanisms that can automatically determine the relative significance between structural and feature information when JAT uses them to compute attention scores. Third, to manifest the contributions of the paper, we perform additional experiments that may further validate the effectiveness of the proposed attention mechanisms. Specifically, we include a much more challenging dataset, i.e., OGBN-Arxiv into our experiment to test the effectiveness of different baselines. We also let JAT utilize different strategies to obtain structural information (e.g., similarity measures, and subspace learning measure) to learn representations. The corresponding results are reported in Table 3 in the revised manuscript. As the table shows, JAT still perform robustly even when it only uses structural correlation computed by some simple measures, such as Cosine and Jaccard similarity. We also let one version of JAT (JAT-SSL) only use structural coefficients to compute attention scores for feature aggregation. Such version of JAT is similar to the ones mentioned by the reviewer that attempt to make a trade-off between spectral information and node features for representation learning. This version of JAT obtains similar results in the corresponding datasets reported by the papers mentioned by the reviewer, but its performance is still a bit worse than the original JAT model. Given such observations, the joint attention mechanisms proposed in this manuscript are the main reasons boosting the performance of the proposed JAT. The generic joint attention framework (its expressive power is verified) proposed in the paper allows JAT to combine with any effective measure/approaches to computing/learning structural coefficients in the graph data. The main contribution of our paper is to provide novel ways (effectively considering structural information) to compute feature aggregations for message passing in attention-based GNNs, rather than to push spectral information into spatial GNNs. The flexibility of JAT allows it to potentially be combined with many existing methods, which may or may not be spectral based. Structural coefficient learned from graph subspace is one effective alternative for JAT and never considered by other attention based GNNs. For the above reasons, we did not take a deep investigation in the original manuscript on subspace/spectral learning and mainly investigated different GNNs from a model perspective. To further distinguish the proposed JAT from other related works, including the ones mentioned by the reviewer, we perform a detailed discussion in Section 2.4 and Appendix E of the revised manuscript.

---

> > > ### Author Response · Authors · 2020-11-24
> > > **Author response to Reviewer 2 Part 3**
> > >
> > > Q3: There is a third issue which relates to some limitations of the approach which are more or less obfuscated in the paper. The work focuses on node classification tasks, ie. where data comprises a single network. Why is this? Is it because the method cannot handle a dataset of graphs of varying topology (e.g. as in graph classification)?
> > >
> > > Response to Q3: Theoretically, JAT can work in the inductive case where the training and testing structures are different. As a message-passing GNN for learning representations in graphs, JAT can learn the node-level representations through multi-layer aggregation of node features. These learned representations can be used in inductive learning tasks by further performing an effective readout operation (readout function). As the main contribution of this paper is to propose novel joint attention mechanisms to guide the attention-based GNNs to pay more attention on those neighbors that are structurally and feature correlated, we mainly use transductive learning tasks to test the effectiveness of JAT. In future, we will further test the effectiveness of JAT in graph classification tasks by combing JAT with effective readout functions which either exist or are proposed.
> > >
> > > Q4: Similarly, why (in Section 2.1) the model is restricted to work with only binary node features? Is this limitation purely notational or does it affect the model? The manuscript should be very clear in stating the limitations of the proposed approach, if any, and what bit of the model is possibly introducing such constraints.
> > >
> > > Response to Q4: The mentioned issue has been addressed in the revised manuscript. The limitations of the current JAT model are discussed in the revised manuscript (Appendix E). Specifically, we point out in the revised manuscript that, we assume the input features are in countable space, as the proof of the expressive power of the proposed model follows the assumption of GIN paper. For uncountable feature space, e.g. those real-valued features, some of the statement in this paper may not always hold. We will try to propose novel joint attention mechanisms which are also injective in the uncountable feature space in our future works.
> > >
> > > Q5: In the empirical analysis, Figure 2.c should also report the confidence intervals, as I am not convinced that the difference “with and without ad” is statistically significant.
> > >
> > > Response to Q5: As suggested, the mentioned figure has been redrawn accordingly. Specifically, the Z scores related to the performance differences between JAT and JAT without ad in Cora , Citeseer, and Pubmed are 2.469, 6.957, and 5.031, respectively. This means the proposed automatic determination strategy may significantly improve the performance on transductive learning.
> > >
> > > Q6: The denominator in (5) can be simplified as it is a summation over two elements.
> > >
> > > Response: The mentioned Equation has been rewritten using a simplified form.

---

### Author Response · Authors · 2020-11-11
**Sincere gratitude to all the reviewers**

The authors must express their sincere gratitude to all the reviewers for the careful review and constructive comments. We will try our best to address all the raised points and appropriately revise our paper as soon as possible.

---

### Author Response · Authors · 2020-11-25
**Brief summary of the changes in the revised manuscript**

The authors express their sincere gratitude again to the reviewers for raising so many constructive comments. Briefly speaking, we have made the following changes in the revised manuscript following your suggestions. For details, please see the response to all the reviewers.

1. More datasets (OGBN-Arxiv) have been used and more ablation studies (JATs using different strateties to learn the structural correlations to perform transductive learning tasks) have been performed in the revised manuscript. The redesigned experimental scenario may better validate the effectiveness of JATs.

2. All the suggested related works have been discussed in the revised manuscript.

3. Further discussions have been performed in both main manuscript and appendix to manifest the contributions of the paper, distinguish the proposed JATs from related works, and point out the limitations of the proposed approach.

4. All the confusing concepts mentioned by the reviewers have been explained, and all the figures that are unclear have been redrawn.

We thank you again for your careful review!

---

### Decision · Program_Chairs · 2021-01-07
**Final Decision**

**Decision:**

Reject

**Comment:**

One referee recommends acceptance, while three referees recommend rejection. All referees agree that augmenting GAT with structural information is an interesting direction to explore; however, they raised concerns about the empirical validation of the method, the related work covered, as well as the discussion of insights such as the method's limitations. The rebuttal addresses R2's concerns by better positioning the work w.r.t the literature and by discussing the method's potential limitations. The rebuttal also covers R1 and R3's comments on scalability and complexity of the proposed approach. However, the rebuttal only partially addresses the evaluation concerns of the reviewers. After discussion, all reviewers agree that further work should be devoted to remedy this. I agree with their assessment and hence must reject. In particular, I would strongly recommend following the referees' suggestions and consider incorporating experiments on additional OGB datasets, including a GAT-based comparison to those (and on OGB-arxiv), and eventually toning down their claims.